# Theoretical Investigations and Practical Enhancements on Tail Task Risk Minimization in Meta Learning

**Yiqin Lv**    **Qi Wang***    **Dong Liang***    **Zheng Xie***
College of Science, National University of Defense Technology
Changsha, China
Email to: {lvyiqin98,wangqi15,dongliangnudt,xiezheng81}@nudt.edu.cn

## Abstract

Meta learning is a promising paradigm in the era of large models and task distributional robustness has become an indispensable consideration in real-world scenarios. Recent advances have examined the effectiveness of tail task risk minimization in fast adaptation robustness improvement [1]. This work contributes to more theoretical investigations and practical enhancements in the field. Specifically, we reduce the distributionally robust strategy to a max-min optimization problem, constitute the Stackelberg equilibrium as the solution concept, and estimate the convergence rate. In the presence of tail risk, we further derive the generalization bound, establish connections with estimated quantiles, and practically improve the studied strategy. Accordingly, extensive evaluations demonstrate the significance of our proposal and its scalability to multimodal large models in boosting robustness.

## 1   Introduction

The past few years have witnessed a surge of research interest in meta learning due to its great potential in the academia and industry [2–5]. By leveraging previous experience, such a learning paradigm can extract knowledge as priors and empower learning models with adaptability to unseen tasks from a few examples [6].

Nevertheless, the investigation of the robustness needs to be more comprehensive from the task distribution perspective. In particular, the recently developed large models heavily rely on the few-shot learning capability and demand robustness of prediction in risk-sensitive scenarios [7]. For example, when the GPT-like dialogue generation system [8–10] comes into medical consultancy domains, imprecise answers can cause catastrophic consequences to patients, families, and even societies in real-world scenarios. In light of these considerations, it is desirable to watch adaptation differences across tasks when deploying meta learning models and promote task robustness study for meeting substantial practical demands.

Recently, Wang et al. [1] proposes to increase task distributional robustness via employing the tail risk minimization principle [11] for meta learning. In circumventing the optimization intractability in the presence of nonconvex risk functions, a two-stage optimization strategy is adopted as the heuristic to solve the problem. In brief, the strategy consists of two phases in iteration, respectively: (i) estimating the risk quantile $VaR_\alpha$ [11] with the crude Monte Carlo method [12] in the task space; (ii) updating the meta learning model parameters from the screened subset of tasks. Such a strategy is simple in implementation, with an improvement guarantee under certain conditions, and empirically shows improved robustness when faced with task distributional shifts. Despite these advances, there remain several unresolved theoretical or practical issues in the field.

---

*Correspondence Authors.

38th Conference on Neural Information Processing Systems (NeurIPS 2024).

**Existing limitations.** This paper also works on the robustness of fast adaptation in the task space and tries to fill gaps in [1]. Theoretically, we notice that in [1] (i) there constitutes no notion of solutions, (ii) it lacks an algorithmic understanding of the two-stage optimization strategy, (iii) the analysis on generalization capability is ignored in the tail risk of tasks. Empirically, the use of the crude Monte Carlo might be less efficient in quantile estimates and suffers from a higher approximation error of the $VaR_\alpha$, degrading the adaptation robustness. These bottlenecks may weaken the versatility of the two-stage optimization strategy's use in practice and require more understanding before deployment.

**Primary contributions.** In response to the above-mentioned concerns, we propose translating the two-stage optimization strategy for distributionally robust meta learning [1] into a max-min optimization problem [13]. Intrinsically, this work models the optimization steps as a Stackelberg game, and task selection and the sub-gradient optimizer work as the leader and follower players in decision-making, respectively. The theoretical understanding is from two aspects:

1. We constitute the local Stackelberg equilibrium as a solution concept, estimate the convergence rate, and characterize the asymptotic behavior in learning dynamics.

2. We derive the generalization bound in the presence of the tail task risk, which connects quantile estimates with fast adaptation capability in unseen tasks.

Meanwhile, the empirical influence of $VaR_\alpha$ estimators is examined, and we advance meta learners' robustness by comprising more accurate quantile estimators.

## 2 Literature Review

### 2.1 Meta Learning

Meta learning, or *learning to learn*, is an increasingly popular paradigm to distill knowledge from prior experience to unseen scenarios with a few examples [6]. Various meta learning methods have emerged in the past decade, and this section overviews some dominant families.

The context-based methods mainly use the encoder-decoder structure and represent tasks by latent variables. Typical ones are in the form of the conditional exchangeable stochastic processes and learn function distributions, such as neural processes [14], conditional neural processes [15] and their extensions [16–24]. The optimization-based approaches seek the optimal meta initialization of model parameters and update models from a few examples. Widely known are model agnostic meta learning [25] and related variants [26–29], such as MetaCurvature [30], which learns curvature information and transforms gradients in the inner-loop optimization. The metrics-based methods represent tasks in geometry and perform well in few-shot image classification [31–33]. For example, MetaOptNet [34] proposes to learn embeddings under a linear classifier and achieve SOTA few-shot classification performance. There also exist other methods, e.g., hyper-networks [35, 36], memory-augmented networks [37] and recurrent models [38].

### 2.2 Robustness & Generalization

The robustness concept in meta learning attracts recent attention, particularly when deploying large models in real-world scenarios. Admittedly, previous literature works have investigated the scenarios where the meta dataset's input is corrupted [39, 40] or the model parameter is perturbed [29]. Studies regarding the fast adaptation robustness in task distribution remain limited. Wang et al. [41] explicitly generates task distribution for robust adaptation. Collins et al. [42] employs the worst-case optimization for promoting MAML's robustness to extreme worst cases. With the help of tail risk minimization, Wang et al. [1] proposes two-stage optimization strategies to robustify the fast adaptation. This work centers around [1] but stresses more theoretical understandings and performance improvement points.

As for generalization capability, there are a couple of works in meta learning. Chen et al. [43] exploits the information theory to derive the bound for MAML's like methods. From the data splitting perspective, Bai et al. [44] formulates the theoretical foundation and connects it to optimality. In [45], an average risk bound is constructed with the bias for improving performance. Importantly, prior work [1] ignores the generalization analysis, and meta learner's generalization in tail risk cases has not been studied in the literature.

# 3 Preliminaries

**General notations.** Let $p(\tau)$ be the task distribution in meta learning. We respectively express the task space and the model parameter space as $\Omega_\tau$ and $\Theta$. We denote the complete task set by $\mathcal{T}$ and refer to $\mathfrak{D}_\tau$ as the meta dataset.

For instance, $\mathfrak{D}_\tau$ comprises a collection of data points $\{(x_i, y_i)\}_{i=1}^{n+m}$ in regression. $\mathfrak{D}_\tau$ is ususally prepared into the support set $\mathfrak{D}_\tau^S$ for skill transfer and the query set $\mathfrak{D}_\tau^Q$ to assess adaptation performance. Take the conditional neural process [15] as an example, $\mathfrak{D}_\tau^S = \{(x_i, y_i)\}_{i=1}^{n}$ works for task representation with $\mathfrak{D}_\tau^Q = \{(x_i, y_i)\}_{i=1}^{n+m}$ the all data points to fit in regression.

The meta risk function corresponds to a map $\ell : \mathfrak{D}_\tau \times \Theta \mapsto \mathbb{R}^+$, evaluating fast adaptation performance. Given $p(\tau)$ and meta learning model parameters $\theta$, we can induce the cumulative distribution of the meta risk function value in the real space as $F_\ell(l; \theta) := \mathbb{P}(\{\ell(\mathfrak{D}_\tau^Q, \mathfrak{D}_\tau^S; \theta) \leq l; \tau \in \mathcal{T}, l \in \mathbb{R}^+\})$, but there is no explicit parameterized form for $F_\ell$ in practice as $F_\ell$ is $\theta$-dependent.

When it comes to the tail risk minimization, we commonly use the conditional value-at-risk ($\text{CVaR}_\alpha$) with the probability threshold $\alpha \in [0, 1)$. The quantile of our interest is called the value-at-risk ($\text{VaR}_\alpha$) [11] with the definition: $\text{VaR}_\alpha[\ell(\mathcal{T}, \theta)] = \inf_{l \in \mathbb{R}^+}\{l | F_\ell(l; \theta) \geq \alpha, \tau \in \mathcal{T}\}$. The resulting normalized cumulative distribution $F_\ell^\alpha(l; \theta)$ is defined as:

$$F_\ell^\alpha(l; \theta) = \begin{cases} 0, & l < \text{VaR}_\alpha[\ell(\mathcal{T}, \theta)] \\ \frac{F_\ell(l;\theta)-\alpha}{1-\alpha}, & l \geq \text{VaR}_\alpha[\ell(\mathcal{T}, \theta)]. \end{cases}$$

$\forall \theta \in \Theta$, the meta learning operator $\mathcal{M}_\theta$ defines: $\mathcal{M}_\theta : \tau \mapsto \ell(\mathfrak{D}_\tau^Q, \mathfrak{D}_\tau^S; \theta)$. Accordingly, the tail risk task subspace $\Omega_{\alpha, \tau} := \bigcup_{\ell \geq \text{VaR}_\alpha[\ell(\mathcal{T}, \theta)]}\left[\mathcal{M}_\theta^{-1}(\ell)\right]$, with the task distribution constrained in $\Omega_{\alpha, \tau}$ by $p_\alpha(\tau; \theta)$. Please refer to Fig. 7 for illustrations of risk concepts.

**Assumption 1.** *To proceed, we retain most assumptions from [1] for theoretical analysis, including:*

1. *The meta risk function $\ell(\mathfrak{D}_\tau^Q, \mathfrak{D}_\tau^S; \theta)$ is $\beta_\tau$-Lipschitz continuous w.r.t. $\theta$;*

2. *The cumulative distribution $F_\ell(l; \theta)$ is $\beta_\ell$-Lipschitz continuous w.r.t. $l$, and the normalized density function $p_\alpha(\tau; \theta)$ is $\beta_\theta$-Lipschitz continuous w.r.t. $\theta$;*

3. *For arbitrary valid $\theta \in \Theta$ and corresponding $p_\alpha(\tau; \theta)$, $\ell(\mathfrak{D}_\tau^Q, \mathfrak{D}_\tau^S; \theta)$ is bounded: $\sup_{\tau \in \Omega_{\alpha, \tau}} \ell(\mathfrak{D}_\tau^Q, \mathfrak{D}_\tau^S; \theta) \leq \mathcal{L}_{\max}$.*

## 3.1 Risk Minimization Principles

This subsection revisits commonly used risk minimization principles in the meta learning field.

**Expected risk minimization.** The standard principle is the expected/empirical risk minimization originated from statistical learning theory [46]. It minimizes meta risk based on the sampling chance of tasks from the original task distribution:

$$\min_{\theta \in \Theta} \mathcal{E}(\theta) := \mathbb{E}_{p(\tau)}\left[\ell(\mathfrak{D}_\tau^Q, \mathfrak{D}_\tau^S; \theta)\right]. \tag{1}$$

**Worst-case risk minimization.** Noticing that the worst fast adaptation can be disastrous in some risk sensitive scenarios, Collins et al. [42] proposes to conduct the worst-case optimization in meta learning:

$$\min_{\theta \in \Theta} \max_{\tau \in \mathcal{T}} \mathcal{E}_{\text{w}}(\theta) := \ell(\mathfrak{D}_\tau^Q, \mathfrak{D}_\tau^S; \theta). \tag{2}$$

However, as observed from experiments in [42], such a principle inevitably sacrifices too much average performance for gains of worst-case robustness. Meanwhile, it requires a couple of implementation tricks and specialized algorithms in stabilizing optimization.

**Expected tail risk minimization ($\text{CVaR}_\alpha$).** To balance the average performance and the worst-case performance, Wang et al. [1] minimizes the expected tail risk, or equivalently $\text{CVaR}_\alpha$ risk measure:

$$\min_{\theta \in \Theta} \mathcal{E}_\alpha(\theta) := \mathbb{E}_{p_\alpha(\tau;\theta)}\left[\ell(\mathfrak{D}_\tau^Q, \mathfrak{D}_\tau^S; \theta)\right]. \tag{3}$$

Due to no closed form of $p_\alpha(\tau;\theta)$, Wang et al. [1] introduces a slack variable $\xi \in \mathbb{R}$ and reformulates the objective as follows:

$$\min_{\theta \in \Theta, \xi \in \mathbb{R}} \mathcal{E}_\alpha(\theta, \xi) := \frac{1}{1-\alpha} \int_\alpha^1 v_\beta d\beta = \xi + \frac{1}{1-\alpha} \mathbb{E}_{p(\tau)} \left[ \left[ \ell(\mathfrak{D}_\tau^Q, \mathfrak{D}_\tau^S; \theta) - \xi \right]^+ \right], \quad (4)$$

where $v_\beta := F_\ell^{-1}(\beta)$ denotes the quantile statistics and $\left[ \ell(\mathfrak{D}_\tau^Q, \mathfrak{D}_\tau^S; \theta) - \xi \right]^+ :=$ $\max\{\ell(\mathfrak{D}_\tau^Q, \mathfrak{D}_\tau^S; \theta) - \xi, 0\}$ is the hinge risk.

The optimization objective involves the integral of quantiles in a continuous interval $(\alpha, 1]$, which is intractable to precisely parameterize with neural networks. The form in Eq. (4) utilizes the duality trick [11], enabling tractable sampling from the complete task space.

## 3.2 Examples & Two-stage Heuristic Strategies

Before delving deeper into the theoretical issues, we first present DR-MAML [1] as an instantiation to explain the expected tail risk minimization.

**Example 1** (DR-MAML [1]). *Given $p(\tau)$ and vanilla MAML [25], the distributionally robust MAML within $CVaR_\alpha$ can be written as a bi-level optimization problem:*

$$\min_{\substack{\theta \in \Theta \\ \xi \in \mathbb{R}}} \xi + \frac{1}{1-\alpha} \mathbb{E}_{p(\tau)} \left[ \left[ \ell(\mathfrak{D}_\tau^Q; \theta - \lambda \nabla_\theta \ell(\mathfrak{D}_\tau^S; \theta)) - \xi \right]^+ \right], \quad (5)$$

*where the gradient update w.r.t. the support set $\nabla_\theta \ell(\mathfrak{D}_\tau^S; \theta)$ indicates the inner loop with a learning rate $\lambda$. The outer loop executes the gradient updates w.r.t. Eq. (5) and seeks the robust meta initialization in the parameter space.*

**Two-stage optimization strategies.** Without loss of generality, we further detail the computational pipelines of Example 1 with two-stage optimization strategies. Note that MAML [25] is an optimization-based meta learning method, and the implementation is to execute the sub-gradient descent over a batch of tasks when updating the meta initialization $\theta^{\text{meta}}$:

$$\theta_t^{\tau_i} = \theta_t^{\text{meta}} - \lambda_1 \nabla_\theta \ell(\mathfrak{D}_{\tau_i}^S; \theta), \ i = 1, \dots, \mathcal{B} \quad (6a)$$

$$\hat{\xi} = \hat{F}_{\text{MC-}\mathcal{B}}^{-1}(\alpha), \quad (6b)$$

$$\delta(\tau_i) = 1[\ell(\mathfrak{D}_{\tau_i}^Q; \theta_t^{\tau_i}) \geq \hat{\xi}], \ i = 1, \dots, \mathcal{B} \quad (6c)$$

$$\theta_{t+1}^{\text{meta}} \leftarrow \theta_t^{\text{meta}} - \lambda_2 \left[ \sum_{i=1}^{\mathcal{B}} \nabla_\theta [\delta(\tau_i) \cdot \ell(\mathfrak{D}_{\tau_i}^Q; \theta_t^{\tau_i})] \right]. \quad (6d)$$

Here, $\lambda_1$ and $\lambda_2$ are the inner loop and the outer loop learning rates, and the subscript $t$ records the iteration number, with $\delta(\tau_i)$ the indicator variable. $\hat{F}_{\text{MC-}\mathcal{B}}$ is the empirical distribution with $\mathcal{B}$ Monte Carlo task samples. $\delta(\tau_i) = 1$ indicates the meta risk $\ell(\mathfrak{D}_{\tau_i}^Q; \theta_t^{\tau_i})$ after fast adaptation falls into the defined tail risk region, otherwise $\delta(\tau_i) = 0$.

Throughout optimizing DR-MAML, **Stage-I** includes the fast adaptation *w.r.t.* individual task in Eq. (6a), and the quantile estimate in Eq. (6b). **Stage-II** applies the sub-gradient updates to the model parameters in Eq. (6c)/(6d). These two stages repeat until convergence is achieved.

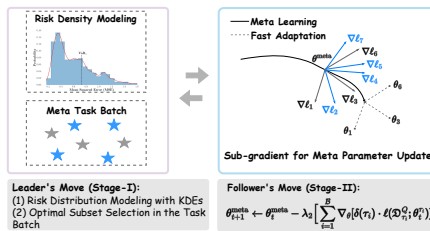

Figure 1: **Illustration of optimization stages in distributionally robust meta learning from a Stackelberg game.** Given the DR-MAML example, the pipeline can be interpreted as bi-level optimization: the leader's move for characterizing tail task risk and the follower's move for robust fast adaptation.

## 4 Theoretical Investigations

This section presents theoretical insights into two-stage optimization strategies. We perform analysis from the algorithmic convergence, the asymptotic tail risk robustness, and the cross-task generalization capability in meta learning.

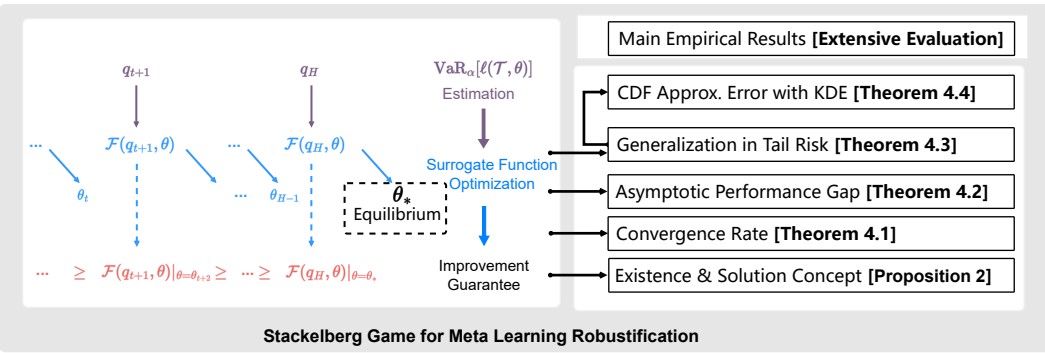

Figure 2: **The sketch of theoretical and empirical contributions in two-stage robust strategies.** On the left side is the two-stage distributionally robust strategy [1]. The contributed theoretical understanding is right-down, with the right-up the empirical improvement. Arrows show connections between components.

## 4.1 Distributionally Robust Meta Learning as a Stackelberg Game

Implementing the two-stage optimization strategy in meta learning requires first specifying the stages' order. The default is the minimization of the risk measure *w.r.t.* the parameter space after the maximization of the risk measure *w.r.t.* the task subspace. Hence, we propose to connect it to max-min optimization [13] and the Stackelberg game [47].

**Max-min optimization.** With the pre-assigned decision-making orders, the studied problem can be characterized as:

$$\max_{q(\tau) \in \mathcal{Q}_\alpha} \min_{\theta \in \Theta} \mathcal{F}(q, \theta) := \mathbb{E}_{q(\tau)}\Big[\ell(\mathfrak{D}_\tau^Q, \mathfrak{D}_\tau^S; \theta)\Big], \tag{7}$$

where $\mathcal{Q}_\alpha := \{q(\tau)|\mathcal{T}_q \subseteq \mathcal{T}, \int_{\tau \in \mathcal{T}_q} p(\tau)d\tau = 1 - \alpha\}$ constitutes a collection of uncertainty sets [48] over task subspace $\mathcal{T}_q$, and $q(\tau)$ is the normalized probability density over the task subspace. Note that in the expected tail risk minimization principle, there is no closed form of optimization objective Eq. (4) as the tail risk is $\theta$-dependent. It is approximately interpreted as the max-min optimization when applied to the distribution over the uncertainty set $\mathcal{Q}_\alpha$.

**Proposition 1.** *The uncertainty set $\mathcal{Q}_\alpha$ is convex and compact in terms of probability measures.*

Practical optimization is achieved via mini-batch gradient estimates and sub-gradient updates with the task size $\mathcal{B}$ in [1]; the feasible subsets correspond to all combinations of size $\lceil \mathcal{B} * (1 - \alpha) \rceil$. Also, Eq. (7) is non-differentiable *w.r.t.* $q(\tau)$, leaving previous approaches [49–52] unavailable in practice.

**Stackelberg game & best responses.** The example computational pipelines in Eq. (6) can be understood as approximately solving a stochastic two-player zero-sum Stackelberg game. Mathematically, such a game referred to as $\mathcal{SG}$ can be depicted as $\mathcal{SG} := \langle \mathcal{P}_L, \mathcal{P}_F; \{q \in \mathcal{Q}_\alpha\}, \{\theta \in \Theta\}; \mathcal{F}(q, \theta)\rangle$.

Moreover, we translate the two-stage optimization as decisions made by two competitors, which are illustrated in Fig. 1. The maximization operator executes in the task space, corresponding to the leader $\mathcal{P}_L$ in $\mathcal{SG}$ with the utility function $\mathcal{F}(q, \theta)$. The follower $\mathcal{P}_F$ attempts to execute sub-gradient updates over the meta learners' parameters via maximizing $-\mathcal{F}(q, \theta)$.

The two players compete to maximize separate utility functions in $\mathcal{SG}$, which can be characterized as:

$$\mathcal{SG}: \underbrace{q_t = \arg\max_{q \in \mathcal{Q}_\alpha} \mathbb{E}_q\Big[\ell(\mathfrak{D}_\tau^Q, \mathfrak{D}_\tau^S; \theta_t)\Big]}_{\textbf{Leader Player}}, \quad \underbrace{\theta_{t+1} = \arg\min_{\theta \in \Theta} \mathbb{E}_{q_t}\Big[\ell(\mathfrak{D}_\tau^Q, \mathfrak{D}_\tau^S; \theta)\Big]}_{\textbf{Follower Player}}, \tag{8}$$

where the leader player $\mathcal{P}_L$ specifies the worst case combinations from the uncertainty set $\mathcal{Q}_\alpha$, and the follower $\mathcal{P}_F$ reacts to the resulting normalized tail risk for increasing fast adaptation robustness.

It is worth noting that the update rules in Eq. (8) are also called *best responses* of players in game theory. The above procedures can be deemed the bi-level optimization [53] since the update of the meta learner implicitly depends on the leader's last time decision.

## 4.2 Solution Concept & Properties

The improvement guarantee has been demonstrated when employing two-stage optimization strategies for minimizing the tail risk in [1]. Furthermore, we claim that under certain conditions, there converges to a solution for the proposed Stackelberg game $\mathcal{SG}$. The sufficient evidence is:

1. The two-stage optimization [1] results in a monotonic sequence:

$$Model\ Updates : \cdots \mapsto \{q_{t-1}, \theta_t\} \mapsto \{q_t, \theta_{t+1}\} \mapsto \cdots \tag{9a}$$

$$Monotonic\ Improvement : \cdots \geq \mathcal{F}(q_{t-1}, \theta_t) \geq \mathcal{F}(q_t, \theta_{t+1}) \geq \cdots; \tag{9b}$$

2. As $\ell \leq \mathcal{L}_{\max}$, the objective $\mathbb{E}_q \left[ \ell(\mathfrak{D}_\tau^Q, \mathfrak{D}_\tau^S; \theta) \right] \leq \mathcal{L}_{\max}$ naturally holds $\forall q \in \mathcal{Q}_\alpha$ and $\theta \in \Theta$.

Built on the boundness of risk functions and the theorem of improvement guarantee, such an optimization process can finally converge [54]. Then, a crucial question arises concerning the obtained solution: ***What is the notion of the convergence point in the game?***

To answer this question, we need to formulate the corresponding solution concept in $\mathcal{SG}$. Here, the global Stackelberg equilibrium is introduced as follows.

**Definition 1** (Global Stackelberg Equilibrium). Let $(q_*, \theta_*) \in \mathcal{Q}_\alpha \times \Theta$ be the solution. With the leader $q_* \in \mathcal{Q}_\alpha$ and the follower $\theta_* \in \Theta$, $(q_*, \theta_*)$ is called a *global Stackelberg equilibrium* if the following inequalities are satisfied, $\forall q \in \mathcal{Q}_\alpha$ and $\forall \theta \in \Theta$,

$$\inf_{\theta' \in \Theta} \mathcal{F}(q, \theta') \leq \mathcal{F}(q_*, \theta_*) \leq \mathcal{F}(q_*, \theta).$$

**Proposition 2** (Existence of Equilibrium). *Given the Assumption 1, there always exists the global Stackelberg equilibrium as the Definition 1 for the studied $\mathcal{SG}$.*

Nevertheless, the existence of the global Stackelberg equilibrium can be guaranteed; it is NP-hard to obtain the equilibrium with existing optimization techniques. The same as that in [55], we turn to the local Stackelberg equilibrium as the Definition 2, where the notion of the local Stackelberg game is restricted in a neighborhood $\mathcal{Q}'_\alpha \times \Theta'$ in strategies.

**Definition 2** (Local Stackelberg Equilibrium). Let $(q_*, \theta_*) \in \mathcal{Q}_\alpha \times \Theta$ be the solution. With the leader $q_* \in \mathcal{Q}_\alpha$ and the follower $\theta_* \in \Theta$, $(q_*, \theta_*)$ is called a *local Stackelberg equilibrium* for the leader if the following inequalities hold, $\forall q \in \mathcal{Q}'_\alpha$,

$$\inf_{\theta \in \mathcal{S}_{\Theta'}(q_*)} \mathcal{F}(q_*, \theta) \geq \inf_{\theta \in \mathcal{S}_{\Theta'}(q)} \mathcal{F}(q, \theta), \text{where } \mathcal{S}_{\Theta'}(q) := \{\bar{\theta} \in \Theta' | \mathcal{F}(q, \bar{\theta}) \leq \mathcal{F}(q, \theta), \forall \theta \in \Theta'\}.$$

The nature of nonconvex programming comprises the above local optimum, and we introduce concepts below for further analysis. It can be validated that $\mathcal{F}(q, \theta)$ is a quasi-concave function *w.r.t.* $q$, meaning that for any positive number $l \in \mathbb{R}_+$, the set $\{q | q \in \mathcal{Q}_\alpha, \mathcal{F}(q, \theta) > l\}$ is convex in $\mathcal{Q}_\alpha$. As a result, we deduce that there exists an implicit function $h(\cdot) : \Theta \to \mathcal{Q}_\alpha$ such that the condition holds $h(\theta) = q$ with $q = \arg\max_{\bar{q} \in \mathcal{Q}_\alpha} \mathcal{F}(\bar{q}, \theta)$. For the implicit function $h$, along with $\nabla_\theta \mathcal{F}(q, \theta)$, we make the Assumption below.

**Assumption 2.** *The implicit function $h(\cdot)$ is $\beta_h$-Lipschitz continuous w.r.t. $\theta \in \Theta$, and $\nabla_\theta \mathcal{F}(q, \theta)$ is $\beta_q$-Lipschitz continuous w.r.t. $q \in \mathcal{Q}_\alpha$.*

## 4.3 Convergence Rate & Generalization Bound

*Learning to learn* scales with the number of tasks, but the optimization process is computationally expensive [56–60], particularly when large language models are meta learners [3, 61, 62]. In training distributionally robust meta learners, estimating the convergence rate allows monitoring of the convergence and designing early stopping criteria to reach a desirable performance, reducing computational burdens [63]. Consequently, we turn to another question regarding the solution concept: ***What is the convergence rate of the two-stage optimization algorithm?***

The runtime complexity for the leader's move can be easily estimated from subset selection, while the analysis for the follower is non-trivial. Under certain conditions, we can derive the following convergence rate theorem, where $\lambda$ is the learning rate in gradient descent *w.r.t.* $\theta$.

**Theorem 4.1** (Convergence Rate for the Second Player). *Let the iteration sequence in optimization be:* $\cdots \mapsto \{q_{t-1}, \theta_t\} \mapsto \{q_t, \theta_{t+1}\} \mapsto \cdots \mapsto \{q_*, \theta_*\}$, *with the converged equilibirum* $(q_*, \theta_*)$. *Under the Assumption 2 and suppose that* $||I - \lambda \nabla^2_{\theta\theta} \mathcal{F}(q_*, \theta_*)||_2 < 1 - \lambda \beta_q \beta_h$, *we can have* $\lim_{t\to\infty} \frac{||\theta_{t+1} - \theta_*||_2}{||\theta_t - \theta_*||_2} \leq 1$, *and the iteration converges with the rate* $(||I - \lambda \nabla^2_{\theta\theta} \mathcal{F}(q_*, \theta_*)||_2 + \lambda \beta_q \beta_h)$ *when $t$ approaches infinity.*

Moreover, after executing the two-stage algorithm $T$ time steps and given learned $\theta_T^{\text{meta}}$, we can establish a bound on the asymptotic performance gap *w.r.t.* $\text{CVaR}_\alpha$ in Theorem 4.2. For expositional clarity, we simplify $\ell(\mathfrak{D}_\tau^Q, \mathfrak{D}_\tau^S; \theta_*)$, $\ell(\mathfrak{D}_\tau^Q, \mathfrak{D}_\tau^S; \theta_T^{\text{meta}})$, $\text{VaR}_\alpha[\ell(\mathcal{T}, \theta_*)]$, and $\text{VaR}_\alpha[\ell(\mathcal{T}, \theta_T^{\text{meta}})]$ as $\ell^*$, $\ell^{\text{meta}}$, $\text{VaR}_\alpha^*$, and $\text{VaR}_\alpha^{\text{meta}}$, respectively.

**Theorem 4.2** (Asymptotic Performance Gap in Tail Task Risk). *Under the Assumption 1 and given a batch of tasks $\{\tau_i\}_{i=1}^{\mathcal{B}}$, we can have*

$$CVaR_\alpha(\theta_T^{meta}) - CVaR_\alpha(\theta_*) \leq \beta_\tau \|\theta_T^{meta} - \theta_*\| + \frac{VaR_\alpha^*}{1-\alpha}\Big(\mathbb{P}(\mathcal{T}_1) - \mathbb{P}(\mathcal{T}_2)\Big), \tag{10}$$

*where $\mathcal{T}_1 = \{\tau : \ell^* < VaR_\alpha^*, \ell^{meta} \geq VaR_\alpha^{meta}\}, \mathcal{T}_2 = \{\tau : \ell^* \geq VaR_\alpha^*, \ell^{meta} < VaR_\alpha^{meta}\}$.*

For sufficiently large $T$, the first term can be bounded by a small number due to the convergence, and the second term vanishes since $\lim_{T\to\infty} \ell^{\text{meta}} = \ell^*$ and $\lim_{T\to\infty} \text{VaR}_\alpha^{\text{meta}} = \text{VaR}_\alpha^*$, respectively.

Another crucial issue regarding meta learning lies in the fast adaptation capability in unseen cases. This drives us to answer the following question: ***How does the resulting meta learner generalize in the presence of tail task risk?***

To this end, we first define $R(\theta_*) = \mathbb{E}_{p_\alpha(\tau)}[\ell^*]$, $\widehat{R}(\theta_*) = \frac{1}{\mathcal{B}} \sum_{i=1}^{\mathcal{B}} \delta(\tau_i)\ell(\mathfrak{D}_{\tau_i}^Q, \mathfrak{D}_{\tau_i}^S; \theta_*)$, and $\widehat{R}_w(\theta_*) = \frac{1}{\mathcal{B}} \sum_{i=1}^{\mathcal{B}} \frac{p_\alpha(\tau_i)}{p(\tau_i)} \ell(\mathfrak{D}_{\tau_i}^Q, \mathfrak{D}_{\tau_i}^S; \theta_*)$, where $\tau_i \sim p(\tau)$. Also note that the support of $p_\alpha(\tau; \theta_*)$ is within that of $p(\tau)$, namely $\text{supp}(p_\alpha(\tau; \theta_*)) \subseteq \text{supp}(p(\tau))$. Then we can induce Theorem 4.3 *w.r.t.* the tail risk generalization.

**Theorem 4.3** (Generalization Bound in the Tail Risk Cases). *Given a collection of task samples $\{\tau_i\}_{i=1}^{\mathcal{B}}$ and corresponding meta datasets, we can derive the following generalization bound in the presence of tail risk:*

$$R(\theta_*) \leq \widehat{R}(\theta_*) + \sqrt{\frac{2\big(\frac{\alpha}{1-\alpha}\mathcal{L}_{\max}^2 + \mathbb{V}_{\tau_i \sim p_\alpha(\tau)}\big[\ell(\mathfrak{D}_{\tau_i}^Q, \mathfrak{D}_{\tau_i}^S; \theta_*)\big]\big)\ln\big(\frac{1}{\epsilon}\big)}{\mathcal{B}}}$$
$$+ \frac{1}{3(1-\alpha)}\frac{\mathcal{L}_{\max}}{\mathcal{B}}\left(2\ln\left(\frac{1}{\epsilon}\right) + 3\alpha\mathcal{B}\right), \tag{11}$$

*where the inequality holds with probability at least $1 - \epsilon$ and $\epsilon \in (0, 1)$, $\mathbb{V}[\cdot]$ denotes the variance operation, and $\mathcal{L}_{\max}$ is from the Assumption 1.*

In conjunction with the confidence $\epsilon$ and a task batch $\mathcal{B}$ of significant size, Theorem 4.3 reveals the generalization bound given the meta-trained parameter $\theta_*$. It is also associated with the variance $\mathbb{V}_{\tau_i \sim p_\alpha(\tau)}[\ell(\mathfrak{D}_{\tau_i}^Q, \mathfrak{D}_{\tau_i}^S; \theta_*)]$. Besides, we also derive a specific bound in the case of MAML, and details are attached in Appendix Theorem C.1.

### 4.4 Practical Enhancements & Implementations

Theorem 4.3 reveals that an accurate estimate of $\text{VaR}_\alpha$ yields a precise variance (i.e., $\mathbb{V}_{\tau_i \sim p_\alpha(\tau)}[\ell(\mathfrak{D}_{\tau_i}^Q, \mathfrak{D}_{\tau_i}^S; \theta_*)]$), leading to more reliable bounds. Accordingly, this section offers improvements over [1] via utilizing kernel density estimators (KDE) [64] for $\text{VaR}_\alpha$'s estimates. Compared to crude Monte Carlo (MC) methods, KDE can handle arbitrary complex distributions, capture local statistics well, and smoothen the cumulative function in a non-parametric way.

Specifically, we can construct KDE with a batch of task risk values $\{\ell(\mathfrak{D}_{\tau_i}^Q, \mathfrak{D}_{\tau_i}^S; \theta)\}_{i=1}^{\mathcal{B}}$:

$$F_{\ell\text{-KDE}}(l; \theta) = \int_{-\infty}^l \frac{1}{\mathcal{B}h_\ell} \sum_{i=1}^{\mathcal{B}} K\Big(\frac{t - \ell(\mathfrak{D}_{\tau_i}^Q, \mathfrak{D}_{\tau_i}^S; \theta)}{h_\ell}\Big)dt, \tag{12}$$

where $K : \mathbb{R}^d \to \mathbb{R}$ is a kernel function, e.g., the Gaussian kernel, $K(x) = \frac{\exp(-||x||^2/2)}{\int \exp(-||x||^2/2)dx}$, and $h_\ell$ is the smoothing bandwidth. Once the KDE is built, it enables access to the quantile from the cumulative distribution functions or numeric integrals. The following Theorem 4.4 shows that KDE serves as a reliable approximation for VaR$_\alpha$.

**Theorem 4.4.** *Let $F_{\ell\text{-}KDE}^{-1}(\alpha;\theta) = VaR_\alpha^{KDE}[\ell(\mathcal{T},\theta)]$ and $F_\ell^{-1}(\alpha;\theta) = VaR_\alpha[\ell(\mathcal{T},\theta)]$. Suppose that $K(x)$ is lower bounded by a constant, $\forall x$. For any $\epsilon > 0$, with probability at least $1 - \epsilon$, we can have the following bound:*

$$\sup_{\theta \in \Theta} \left( F_{\ell\text{-}KDE}^{-1}(\alpha;\theta) - F_\ell^{-1}(\alpha;\theta) \right) \leq \mathcal{O}\left( \frac{h_\ell}{\sqrt{\mathcal{B} * \log \mathcal{B}}} \right). \tag{13}$$

As implied, one can close the distribution approximation gap by adopting a smaller, more flexible bandwidth. Additionally, KDE models offer a smooth estimate of the cumulative distribution function and require no prior assumptions.

**Remark 1.** In addition to smoothness, flexibility, and distribution agnostic traits, KDE in adoption can enhance the studied method's generalization capability. The crude Monte Carlo used in [1] typically incurs an error of approximately $\mathcal{O}(\frac{1}{\sqrt{\mathcal{B}}})$ in estimating quantiles [65]. In contrast, that of KDE is no more than $\mathcal{O}(\frac{h_\ell}{\sqrt{\mathcal{B} * \log \mathcal{B}}})$ from Theorem 4.4.

## 5 Empirical Findings

Prior sections mainly focus on the theoretical understanding of two-stage distributionally robust strategies. This section conducts extensive experiments on a broader range of benchmarks and examines the improvement tricks, e.g., the use of KDE for quantile estimates, from empirical results.

**Benchmarks & baselines.** We perform experiments on the few-shot regression, system identification, image classification, and meta reinforcement learning, where most of them keep setups the same as prior work [1, 42]. We evaluate the methods from risk minimization principles and corresponding indicators, including expected/empirical risk minimization (Average), worst-case risk minimization (Worst), and tail risk minimization (CVaR$_\alpha$).

MAML mainly works as the base meta learner, and we term the KDE-augmented DR-MAML as DR-MAML+. Then we compare DR-MAML+ with several baselines, including vanilla MAML [25], TR-MAML [42], DRO-MAML [66] and DR-MAML [1].

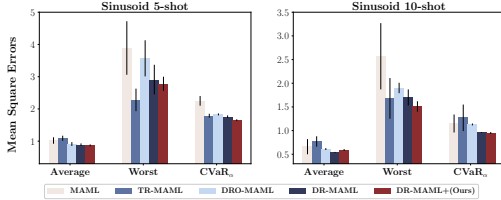

Figure 3: **Meta testing performance in sinusoid regression problems (5 runs).** The charts report testing mean square errors (MSEs) over 490 unseen tasks [42] with $\alpha = 0.7$, where black vertical lines indicate standard error bars.

Figure 4: **Meta testing performance in Pendulum `10-shot` and `20-shot` problems (5 runs).** Reported are testing MSEs over 529 unseen tasks with $\alpha = 0.5$, where black vertical lines indicate standard error bars.

### 5.1 Sinusoid Regression

The goal of the sinusoid regression [25] is to quickly fit an underlying function $f(x) = A\sin(x - B)$ from $K$ randomly sampled data points, and tasks are specified by $(A, B)$. The meta-training and testing setups are the same as that in [1, 42], where many easy functions with a tiny fraction of difficult ones are included in the training.

**Result & analysis.** As illustrated in Fig. 3, we can observe that DR-MAML+ consistently outperforms all baselines across average and CVaR$_\alpha$ indicators in the `5-shot` case. Though the average performance slightly lags behind DR-MAML in the `10-shot` case, DR-MAML+ surpasses other baselines in both the Worst and CVaR$_\alpha$ indicators. This implies that DR-MAML+ exhibits more

robustness in challenging task distributions, e.g., `5-shot` case. Furthermore, the standard error associated with our method is significantly smaller than others, underscoring the stability of DR-MAML+.

## 5.2 System Identification

The system identification corresponds to learning a dynamics model from a few collected transitions in physics systems. Here, we consider the Pendulum system and create diverse dynamical systems by varying its mass $m$ and length $l$, with $(m, l) \sim \mathcal{U}([0.4, 1.6], [0.4, 1.6])$. A random policy collects transitions for meta training, and 10 random transitions work as a support dataset.

**Result & analysis.** Fig. 4 shows no significant difference between `10-shot` and `20-shot` cases. DR-MAML+ dominates the performance across all indicators in both cases. Due to the min-max optimization, TR-MAML behaves well in the worst-case but sacrifices too much average performance. Within the studied strategies, DR-MAML+ exhibits an advantage over DR-MAML regarding $\text{CVaR}_\alpha$.

## 5.3 Few-shot Image Classification

We perform few-shot image classification on the *mini*-ImageNet dataset [67], with the same setup in [42]. The task is a `5-way 1-shot` classification problem. And 64 classes are selected for constructing meta-training tasks, with the remaining 32 classes for meta-testing.

Table 1: **Average `5-way 1-shot` classification accuracies in *mini*-ImageNet with reported standard deviations (3 runs).** With $\alpha = 0.5$, the best results are in bold.

| Method | Eight Meta-Training Tasks | | | Four Meta-Testing Tasks | | |
| --- | --- | --- | --- | --- | --- | --- |
| | Average | Worst | $\text{CVaR}_\alpha$ | Average | Worst | $\text{CVaR}_\alpha$ |
| MAML [25] | 70.1±2.2 | 48.0±4.5 | 63.2±2.6 | 46.6±0.4 | 44.7±0.7 | 44.6±0.7 |
| TR-MAML [42] | 63.2±1.3 | 60.7±1.6 | 62.1±1.2 | 48.5±0.6 | 45.9±0.8 | 46.6±0.5 |
| DRO-MAML [66] | 67.0±0.2 | 56.6±0.4 | 61.6±0.2 | 49.1±0.2 | 46.6±0.1 | 47.2±0.2 |
| DR-MAML [1] | 70.2±0.2 | 63.4±0.2 | 67.2±0.1 | 49.4±0.1 | 47.1±0.1 | 47.5±0.1 |
| DR-MAML+(Ours) | **70.4±0.1** | **63.8±0.2** | **67.5±0.1** | **49.9±0.1** | **47.2±0.1** | **48.1±0.1** |

**Result & analysis.** In Table 1, methods within a two-stage distributionally robust strategy, namely DR-MAML and DR-MAML+, show superiority to others across all indicators in both training and testing scenarios, which is similar to empirical findings in [1]. Interstingly, DR-MAML+ and DR-MAML are comparable in most scenarios, and we attribute this to the small batch size in training, which weakens KDE's quantile approximation advantage.

## 5.4 Meta Reinforcement Learning

Here, we take 2-D point robot navigation as the meta reinforcement learning benchmark in evaluation. The goal is to reach the target destination with the help of a few exploration transitions for fast adaptation, and we retain the setup in MAML [25]. In meta testing, we randomly sample 80 navigation goals and examine methods' navigation performance.

Table 2: **Meta testing returns in point robot navigation (4 runs).** The chart reports average return and $\text{CVaR}_\alpha$ return with $\alpha = 0.5$.

| Method | Average | $\text{CVaR}_\alpha$ |
| --- | --- | --- |
| MAML [25] | -21.1 ± 0.69 | -29.2 ± 1.37 |
| DRO-MAML [66] | -20.9 ± 0.41 | -29.0 ± 0.66 |
| DR-MAML [1] | -19.6 ± 0.49 | -28.9 ± 1.20 |
| DR-MAML+(Ours) | **-19.2± 0.44** | **-28.4± 0.86** |

**Result & analysis.** As reinforcement learning methods fluctuate fiercely in worst-case indicators, we only report Average and $\text{CVaR}_\alpha$ returns in Table 2. We observe that using studied strategies in DR-MAML enhances the returns. DR-MAML+ benefits from a more reliable quantile estimate and achieves superior performance. The application of distributional robustness to reinforcement learning yields improvements in returns.

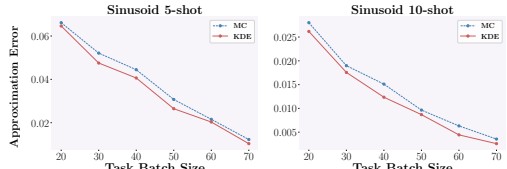

Figure 5: **VaR$_\alpha$ approximation errors with the crude MC and KDE.** We compute the difference between the estimated $\hat{\text{VaR}}_\alpha$ and the Oracle $\text{VaR}_\alpha$ in the absolute value $|\hat{\text{VaR}}_\alpha - \text{VaR}_\alpha|$.

## 5.5 Assessment of Quantile Estimators

With the meta trained model, e.g., DR-MAML+ in sinusoid regression, we collect the testing task risk values with different task batch sizes to estimate the $\text{VaR}_\alpha$ from respectively the crude MC and KDE. As observed from Fig. 5, the $\text{VaR}_\alpha$ approximation error decreases with more tasks, and the KDE produces more accurate estimates with a sharper decreasing trend. The above well verifies the conclusion in Theorem 4.3.

## 5.6 Empricial Result Summarization

Here, we summarize two points from the above empirical results and associated theorems. (i) From Theorem 4.2/4.3 and Fig. 3/4/5: the $\text{VaR}_\alpha$ estimate relates to the reliable generalization bound, and cumulated tiny approximation errors along iterations potentially result in worse equilibrium. (ii) From Theorem 4.3/4.4, Remark 1, Fig. 3, and Table 1/2: with the studied strategy, the KDE is a better choice of task risk distribution modelling than the crude MC in tougher benchmarks, e.g., 5-shot sinusoid regression, meta-testing *mini*-ImageNet classification, and point robot navigation.

## 5.7 Compatibility with Large Models

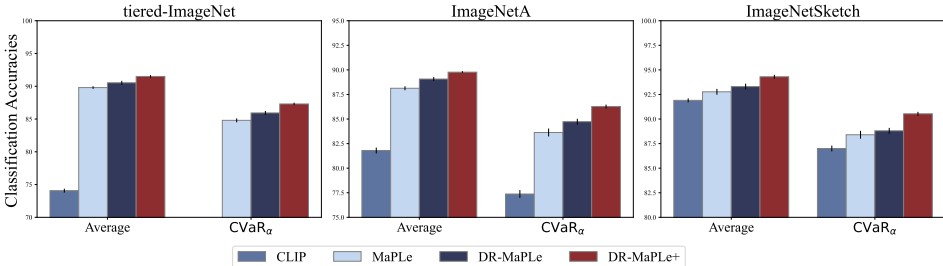

Figure 6: **Meta testing results on** `5-way 1-shot` **classification accuracies with reported standard deviations (3 runs).** The charts respectively report classification accuracies over 150 unseen tasks. We further conduct few-shot image classification experiments in the presence of large model. Note that CLIP [68] exhibits strong zero-shot adaptation capability; hence, we employ "ViT-B/16"-based CLIP as the backbone to enable few-shot learning in the same way as MaPLe with training setup N_CTX = 2 and MAX_EPOCH = 30 [69], scaling to large neural networks in evaluation (See Appendix Section D for details).

**Improved Robustness in Evaluation:** As illustrated in Fig. 6, DR-MaPLe and DR-MaPLe+ consistently outperform baselines across both average and indicators in cases, demonstrating the advantage of the two-stage strategy in enhancing the robustness of few-shot learning. DR-MaPLe+ achieves better results as KDE quantiles are more accurate with large batch sizes. These results confirm the scalability and compatibility of our method on large models.

**Learning Efficiency as Limitations:** In terms of implementation time and memory cost, we retain the setup the same as that in [1]: use the same maximum number of meta gradient updates for all baselines in training processes, which means given $\alpha = 0.5$, the tail risk minimization principle requires double task batches to evaluate and screen sub-batches. It can be seen that both DR-MaPLe and DR-MaPLe+ consume more memories, and the extra training time over MaPLe arises from the evaluation and sub-batch screening in the forward pass. Such additional computations and memory costs raise computational and memory efficiency issues for exchanging extra significant robustness improvement in fast adaptation.

## 6 Conclusion

To conclude, this paper proposes to understand the two-stage distributionally robust strategy from optimization processes, define the convergence solution, and derive the generalization bound in the presence of tail task risk. Extensive experiments validate the studied improvement tricks and reveal more empirical properties of the studied strategy. We leave computational overhead reduction as a promising topic for future exploration in robust fast adaptation.

## Acknowledgement

This work is funded by National Natural Science Foundation of China (NSFC) with the Number # 62306326. We express particular gratitude to friends who guide large model-relevant experiments.

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

# Contents

# A  Quick Guide to This Work

This section mainly includes explanations and clarifications on this work.

## A.1  Technical Comparison in Robust Fast Adaptation

Table 3: **A summary of robust fast adaptation methods.** We take MAML as an example, list related methods, and report their characteristics in literature. We mainly report the statistics according to whether existing literature works include the generalization analysis and convergence analysis. The form of meta learner and the robustness type are generally connected.

| Principle | Meta Learner | Generalization | Convergence | Robustness Type |
|---|---|---|---|---|
| MAML | $\min_{\theta\in\Theta}\mathbb{E}_{p(\tau)}\left[\ell(\mathfrak{D}_\tau^Q,\mathfrak{D}_\tau^S;\theta)\right]$ | ✓ | ✓ | $--$ |
| DRO-MAML [66] | $\max_{q(\tau)\in\mathcal{Q}}\min_{\theta\in\Theta}\mathbb{E}_{q(\tau)}\left[\ell(\mathfrak{D}_\tau^Q,\mathfrak{D}_\tau^S;\theta)\right]$ | ✗ | ✗ | Uncertainty Set (Not tail risk) |
| TR-MAML [42] | $\min_{\theta\in\Theta}\max_{\tau\in\mathcal{T}}\ell(\mathfrak{D}_\tau^Q,\mathfrak{D}_\tau^S;\theta)$ | ✓ | ✓ | Worst-Case Task |
| DR-MAML [1] | $\min_{\theta\in\Theta}\mathbb{E}_{p_\alpha(\tau;\theta)}\left[\ell(\mathfrak{D}_\tau^Q,\mathfrak{D}_\tau^S;\theta)\right]$ | ✗ | ✗ | Tail Task Risk |
| DR-MAML+(Ours) | $\max_{q(\tau)\in\mathcal{Q}_\alpha}\min_{\theta\in\Theta}\mathbb{E}_{q(\tau)}\left[\ell(\mathfrak{D}_\tau^Q,\mathfrak{D}_\tau^S;\theta)\right]$ | ✓ | ✓ | Tail Task Risk |

**Primary differences:** As far as we know, literature work is quite limited regarding fast adaptation robustness in the task space. TR-MAML and DR-MAML are the most recent and typical ones that can handle task distributional shift scenarios well. As reported in Table 3, TR-MAML only focuses on the worst-case, which considers a bit extreme and rarely occurred cases. DRO-MAML is a new baseline, where the uncertainty set $\mathcal{Q}$ is included for robust fast adaptation, hence there exists no theoretical analysis. As for the tail task risk, DR-MAML lacks generalization capability and convergence rate analysis *w.r.t.* the meta learner. The meta learner in DR-MAML+ is a more specific instantiation of that in DR-MAML. We claim that these theoretical understanding is necessary in the presence of the robust fast adaptation due to its potential applications in large models.

**Theoretical and empirical insights:** In comparison, *this work not only contributes to the Stackelberg game for estimates, but also derives the generalization and the asymptotic performance gap in iterations based on a normalized but non-differentiable probability density space.* Note that we lean more focus on theoretical understanding and pursuing SOTA performance is not the ultimate purpose of this work. The connections between different quantile estimators and generalization bound highlighted in Theorem 4.3/4.4 and Remark 1 reveal the theoretical advantage of KDEs over crude Monte Carlo methods. This motivates us to replace crude Monte Carlo with KDEs. Such a replacement as a simple implementation trick is supported by rigorous theoretical analysis. The empirical results align with theoretical understanding.

In terms of improving the studied strategy, *investigations in extensive experiments seem meaningful for practical implementations, and some non-trivial discoveries together with improvement tricks are also reported*, such as the relationship between quantile estimate errors and adaptation robustness, the batch size's influence on several benchmarks, etc.

In this work, the theoretical and empirical parts are connected in an implicit manner. The generalization capability is empirically examined from experimental results, and the performance gap between DR-MAML+ and DR-MAML can be attributed to the difference in generalization bounds. As for the convergence trait and asymptotic performance, the insight might guide the optimization process in training large models, such as early stopping criteria design.

## A.2  Significance of Theoretical Understandings

As pointed out in [3, 61, 62], large language models are few-shot learners. When a large model, such as a large decision-making model in the future, comes into practice, fast adaptation robustness can be a crucial issue as real-world scenarios are indeed risk-sensitive.

This work takes the latest work [1] as an example, and the interest is in the theoretical aspect. Most of the assumptions in this work are from [1]. The baselines are typical and latest, while the benchmarks

cover diverse downstream tasks. In multimodal few-shot image classification experiments, our contributed points help guide the development of large models in terms of training and robustness enhancement. Our investigations also provide insight into robust policy optimization, particularly when safety is one necessary consideration [70–72].

Shapiro et al.'s book [73] is a comprehensive resource that addresses stochastic modeling and optimization methods, but it does not explore solution concepts in game theory or define generalization bounds relevant to meta learning and deep learning. Instead, our work further enriches the stochastic programming theory in meta learning, connects it to the Stackelberg game, and contributes to tail risk generalization bounds, convergence rates, asymptotic properties, and so on. Therefore, the solution concept and the theoretical properties are specific to our meta learning setup, distinctly from the scope covered by [73].

### A.3 Meanings of Indicators and Terms

**Illustration of VaR$_\alpha$, CDF and others:** Fig. 7 illustrates a typical probability distribution, cumulative distribution, and the resulting mean, VaR$_\alpha$, and CVaR$_\alpha$. Given $\alpha \in [0, 1)$, VaR$_\alpha$ is the $\alpha$ quantile of the risk distribution. Specially, VaR$_{0.5}$ coincides with the mean. Upon the definition of VaR$_\alpha$, CVaR$_\alpha$ can be define as CVaR$_\alpha = \mathbb{E}_{p(\tau)}\left[\ell | \ell \geq \text{VaR}_\alpha\right]$. That is, CVaR$_\alpha$ is the expectation of the risks of the $1 - \alpha$ tail of the distribution. Relative to the original probability distribution, CVaR$_\alpha$ can be interpreted as a certain *distribution shift*, which reweighs arbitrary risk exceeding VaR$_\alpha$ up to a coefficient $\frac{1}{1-\alpha}$.

**Meaning of the asymptotic performance gap:** We plot Fig. 8 to display the gap between the CVaR$_\alpha$ value in iterations and that in the convergence. The area difference depicts this gap.

### A.4 Computational Complexity

Analyzing computational complexity across all meta-learning methods is inherently challenging due to the diversity in methodological approaches within the field. Meta-learning encompasses a wide range of techniques, including gradient-based methods, which rely on iterative updates to model parameters, and non-parametric methods, which may instead focus on instance-based learning or kernel-based approaches. Therefore, the space complexity is specific to the meta-learning method, while this work is agnostic to it. Here, we report the computational complexity for the DR-MAML+ as $\mathcal{O}\left(\mathcal{B}(\mathcal{B} - \alpha|\mathcal{M}|)\right)$ while using KDE with the Gaussian kernel, and that of DR-MAML is $\mathcal{O}\left(\mathcal{B}(\log(\mathcal{B}) - \alpha|\mathcal{M}|)\right)$.

### A.5 Broader Impact & Future Extensions

This paper presents work whose goal is to advance the field of robust meta learning. There are many potential societal consequences of our work, which we detail as follows.

The fast adaptation robustness is an urgent concern, particularly in large models and risk-sensitive control. This work provides versatile insights for theoretical analysis and performance improvement in the presence of tail task risk, and future explorations can be decision-making scenarios, such as multi-agent policy optimization [74, 75], and computational/memory cost reduction.

## B Pseudo Algorithms

For a better understanding of the game theoretical optimization, we take DR-MAML+ and DR-CNP+ as examples and include the Pseudo Algorithms 1/2 in this section. Particularly, the algorithms specify the decision-making orders and highlight the use of KDE modules to build task risk value distributions and estimate the quantile.

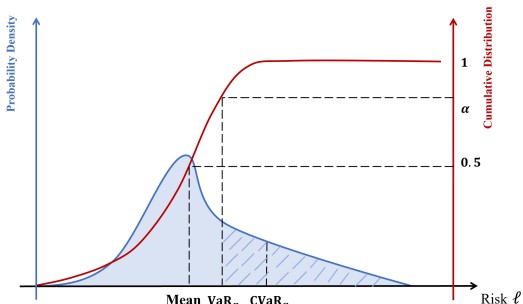

Figure 7: **Diagram of risk concepts in this work.** Here, the $x$-axis is the task risk value in fast adaptation given a specific $\theta$. The shadow-lined region illustrates the tail risk with a probability $1 - \alpha$ in the probability density. The area of the shadow-lined region after $1 - \alpha$ normalization corresponds to the expected tail risk $\text{CVaR}_\alpha$.

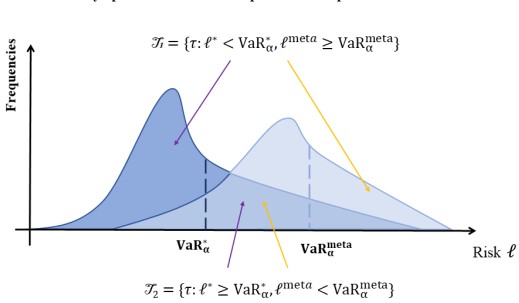

Figure 8: **Illustration of the asymptotic behavior in approximating the equilibrium.** Here, the $x$-axis is the feasible task risk value in fast adaptation. The dark blue region indicates the histogram of the task risk values in the local Stackelberg equilibrium $(q_*, \theta_*)$. The shallow blue region describes the histogram of the task risk values at some iterated point $(q_{T-1}, \theta_T^{\text{meta}})$. The sets $\mathcal{T}_1$ and $\mathcal{T}_2$ respectively collect the tasks resulting the opposite order.

## C  Expressions, Theorems & Proofs

### C.1  Characterization of Optimization Processes

Without loss of generality, we can also express the process of solving the studied Stackelberg game as:

$$\max_{q \in \mathcal{Q}_\alpha} \mathcal{F}(q, \theta_*(q)) \quad \text{s.t. } \theta_*(q) = \arg\min_{\theta \in \Theta} \mathcal{F}(q, \theta) \qquad \textbf{(13a: } \text{Leader's Decision-Making)}$$

$$\min_{\theta \in \Theta} \mathcal{F}(q, \theta), \qquad \textbf{(13b: } \text{Follower's Decision-Making)}$$

where the optimization *w.r.t.* $(q, \theta)$ is the computation of the best responses for two adversarial players. As a bi-level optimization in Eq. (14a)/(14b), the exact solution is intractable to obtain in a theoretical sense, and the two-stage distributionallly robust optimization is a heuristic approach.

**Meaning of the obtained equilibrium.** Here, we can interpret the obtained solution $(q_*, \theta_*)$ from solving Eq. (7) as follows. Given the follower's decision $\theta_*$ and the induced task risk distribution $F_\ell(l; \theta_*)$, the leader cannot further raise a proposal of a task subset with a probability $1 - \alpha$ to degradde the tailed expected performance. And this explains the meaning of robust fast adaptation solution *w.r.t.* the tail task risk.

---
**Algorithm 1:** Meta-training DR-MAML+ as A Stackelberg Game
---
**Input** : Task distribution $p(\tau)$; Confidence level $\alpha$; Task batch size $\mathcal{B}$; Learning rates: $\lambda_1$ and $\lambda_2$.
**Output** : Meta-trained model parameter $\theta$.
Randomly initialize the model parameter $\theta$;
**while** *not converged* **do**
    Sample a batch of tasks $\{\tau_i\}_{i=1}^{\mathcal{B}} \sim p(\tau)$;
    `# The Leader Player's Decision-Making`
    **for** $i = 1$ **to** $\mathcal{B}$ **do**
        `// inner loop via gradient descent as the fast adaptation`
        Evaluate the gradient: $\nabla_\theta \ell(\mathfrak{D}_{\tau_i}^S; \theta)$ in Eq. (5);
        Perform task-specific gradient updates:
        $\theta_i \leftarrow \theta - \lambda_1 \nabla_\theta \ell(\mathfrak{D}_{\tau_i}^S; \theta)$;
    **end**
    `// model the task risk distribution and estimate the quantile`
    Evaluate performance $\mathcal{L}_\mathcal{B} = \{\ell(\mathfrak{D}_{\tau_i}^Q; \theta_i)\}_{i=1}^{\mathcal{B}}$;
    Estimate $\text{VaR}_\alpha[\ell(\mathcal{T}, \theta)]$ and set $\xi = \hat{\xi}_\alpha$ in Eq. (5) with kernel density estimators;
    Screen the subset $\mathcal{L}_{\hat{\mathcal{B}}} = \{\ell(\mathfrak{D}_{\hat{\tau}_i}^Q; \theta_i)\}_{i=1}^K$ with $\hat{\xi}_\alpha$ for meta initialization updates;
    `# The Follower Player's Decision-Making`
    Execute outer loop via gradient descent to increase adaptation robustness:
    $\theta \leftarrow \theta - \lambda_2 \nabla_\theta \sum_{i=1}^K \ell(\mathfrak{D}_{\hat{\tau}_i}^Q; \theta_i)$ in Eq. (5);
**end**
---

---
**Algorithm 2:** Meta Training DR-CNP+ as A Stackelberg Game
---
**Input** : Task distribution $p(\tau)$; Confidence level $\alpha$; Task batch size $\mathcal{B}$; Learning rate $\lambda$.
**Output** : Meta-trained model parameter $\theta$.
Randomly initialize the model parameter $\theta$;
**while** *not converged* **do**
    Sample a batch of tasks $\{\tau_i\}_{i=1}^{\mathcal{B}} \sim p(\tau)$;
    `# The Leader Player's Decision-Making`
    `// model the task risk distribution and estimate the quantile`
    Evaluate performance $\mathcal{L}_\mathcal{B} = \{\ell(\mathfrak{D}_{\tau_i}^Q; z, \theta_i)\}_{i=1}^{\mathcal{B}}$;
    Estimate $\text{VaR}_\alpha[\ell(\mathcal{T}, \theta)] \approx \hat{\xi}_\alpha$ with kernel density estimators;
    Screen the subset $\mathcal{L}_{\hat{\mathcal{B}}} = \{\ell(\mathfrak{D}_{\hat{\tau}_i}^Q; z, \theta)\}_{i=1}^K$ with $\hat{\xi}_\alpha$ for meta initialization updates;
    `# The Follower Player's Decision-Making`
    Execute gradient descent to increase adaptation robustness:
    $\theta \leftarrow \theta - \lambda \nabla_\theta \sum_{i=1}^K \ell(\mathfrak{D}_{\hat{\tau}_i}^Q; z, \theta)$;
**end**
---

### C.2 Assumptions

We list all of the assumptions mentioned in this work. These assumptions further serve the demonstration of propositions and theorems in the main paper.

**Assumption 1.** *To proceed, we retain most assumptions from [1] for theoretical analysis, including:*

1. *The meta risk function $\ell(\mathfrak{D}_\tau^Q, \mathfrak{D}_\tau^S; \theta)$ is $\beta_\tau$-Lipschitz continuous w.r.t. $\theta$;*

2. *The cumulative distribution $F_\ell(l; \theta)$ is $\beta_\ell$-Lipschitz continuous w.r.t. $l$, and the normalized density function $p_\alpha(\tau; \theta)$ is $\beta_\theta$-Lipschitz continuous w.r.t. $\theta$;*

3. *For arbitrary valid $\theta \in \Theta$ and corresponding $p_\alpha(\tau; \theta)$, $\ell(\mathfrak{D}_\tau^Q, \mathfrak{D}_\tau^S; \theta)$ is bounded: $\sup_{\tau \in \Omega_{\alpha,\tau}} \ell(\mathfrak{D}_{\tau_i}^Q, \mathfrak{D}_{\tau_i}^S; \theta) \leq \mathcal{L}_{\max}$.*

**Assumption 2.** *The implicit function $h(\cdot)$ is $\beta_h$-Lipschitz continuous w.r.t. $\theta \in \Theta$, and $\nabla_\theta \mathcal{F}(q, \theta)$ is $\beta_q$-Lipschitz continuous w.r.t. $q \in \mathcal{Q}_\alpha$.*

## C.3 Proof of Proposition 1

**Proposition 1.** *The uncertainty set $\mathcal{Q}_\alpha$ is convex and compact in terms of probability measures.*

**_Proof:_** We firstly focus on the convexity of $\mathcal{Q}_\alpha$. For any $\{q_1 := q_1(\tau), q_2 := q_2(\tau)\} \in \mathcal{Q}_\alpha$, we partition these two task spaces with non-zero sampling probability mass respectively as $\mathcal{T}_1 \cup \mathcal{T}_C$ and $\mathcal{T}_2 \cup \mathcal{T}_C$. As displayed in Fig. 9, $\mathcal{T}_C$ denotes the shared subset task between $q_1$ and $q_2$. Below we show that $\lambda_1 q_1 + \lambda_2 q_2 \in \mathcal{Q}_\alpha$ with $\lambda_1 + \lambda_2 = 1$. This is true because

$$\int_{\tau \in \mathcal{T}_{\lambda_1 q_1 + \lambda_2 q_2}} p(\tau) d\tau \tag{15a}$$

$$= \int_{\tau \notin \mathcal{T}_{q_1} \cup \mathcal{T}_{q_2}} p(\tau) d\tau + \int_{\tau \in \mathcal{T}_C} \left( \lambda_1 p(\tau) + \lambda_2 p(\tau) \right) d\tau + \int_{\tau \in \mathcal{T}_1} \lambda_1 p(\tau) d\tau + \int_{\tau \in \mathcal{T}_2} \lambda_2 p(\tau) d\tau \tag{15b}$$

$$= 0 + \lambda_1 \left( \int_{\tau \in \mathcal{T}_C} p(\tau) d\tau + \int_{\tau \in \mathcal{T}_1} p(\tau) d\tau \right) + \lambda_2 \left( \int_{\tau \in \mathcal{T}_C} p(\tau) d\tau + \int_{\tau \in \mathcal{T}_2} p(\tau) d\tau \right) \tag{15c}$$

$$= \lambda_1 \int_{\tau \in \mathcal{T}_{q_1}} p(\tau) d\tau + \lambda_2 \int_{\tau \in \mathcal{T}_{q_2}} p(\tau) d\tau \tag{15d}$$

$$= 1 - \alpha. \tag{15e}$$

We next demonstrate the compactness of $\mathcal{Q}_\alpha$. The distance between two distributions $\forall \{q_1, q_2\} \in \mathcal{Q}_\alpha$ can be defined as:

$$d_{\mathcal{Q}_\alpha}(q_1, q_2) := \int_{\tau \in \mathcal{T}} \left| q_1(\tau) - q_2(\tau) \right| d\tau.$$

Since $L^1$ space is a Banach space, the compactness is equivalent to the closedness and Boundedness of $\mathcal{Q}_\alpha$. Considering a sequence $\{q_n(\tau) \in \mathcal{Q}_\alpha\}$ with the resulting limitation is $q_*(\tau)$, following the *C*ontrolled Convergence Theorem [76], we know that

$$\lim_{n \to \infty} \int_{\tau \in \mathcal{T}} p_n(\tau) - p_*(\tau) d\tau \le \lim_{n \to \infty} \int_{\tau \in \mathcal{T}} \left| p_n(\tau) - p_*(\tau) \right| d\tau = \lim_{n \to \infty} d_{\mathcal{Q}_\alpha}(q_n, q_*) = 0. \tag{16}$$

Due to the symmetry of the distance, we can have $\lim_{n \to \infty} \int_{\tau \in \mathcal{T}} p_*(\tau) - p_n(\tau) d\tau \le 0$. Thus,

$$\int_{\tau \in \mathcal{T}} p_*(\tau) d\tau = \lim_{n \to \infty} \int_{\tau \in \mathcal{T}} p_n(\tau) d\tau = 1 - \alpha.$$

That is, $p_*(\tau) \in \mathcal{Q}_\alpha$, indicating that $\mathcal{Q}_\alpha$ is a closed set. As the boundedness is clear in the studied problem, this completes the proof of Proposition 1. ∎

## C.4 Proof of Proposition 2

**Proposition 2** (Existence of Equilibrium) *Given the Assumption 1, there always exists the global Stackelberg equilibrium as the Definition 1 for the studied $\mathcal{SG}$.*

**_Proof:_** Note that $\Theta$ is compact as a subspace of the Euclidean space. And it is trivial to see that $\mathcal{F}(q, \theta) := \mathbb{E}_q \left[ \ell(\mathfrak{D}_\tau^Q, \mathfrak{D}_\tau^S; \theta) \right]$ is continuous *w.r.t.* $\theta \in \Theta$ as $\ell$ satisfies the $\beta_\tau$-Lipschitz continuity in the Assumption 1.

Here we need to show the continuity of $\mathcal{F}(q, \theta)$ *w.r.t.* the collection of probability measures or probability functions $\mathcal{Q}_\alpha$. To this end, with $\forall \theta \in \Theta$ fixed, We consider two metric spaces $(\mathcal{Q}_\alpha, d_{\mathcal{Q}_\alpha})$ and $(\mathcal{L}, \mathbb{R}_{\mathcal{L}})$. The map of our interest is $g(q) = \mathcal{F}(q, \cdot) : \mathcal{Q}_\alpha \mapsto \mathcal{L} \subseteq \mathbb{R}^+$.

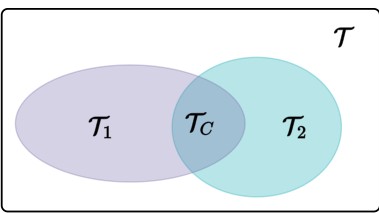

Figure 9: **Partition of the task subspace.** Here we take two probability measure $\{q_1, q_2\} \in \mathcal{Q}_\alpha$ for illustration. $\mathcal{T}_1 \cup \mathcal{T}_C$ and $\mathcal{T}_2 \cup \mathcal{T}_C$ defines the corresponding task subspaces for $q_1$ and $q_2$ with non-zero probability mass in the whole space $\mathcal{T}$.

Naturally, we can have the following inequality:

$$\left| g(q_1) - g(q_2) \right| = \left| \mathbb{E}_{q_1} \left[ \ell(\mathfrak{D}_\tau^Q, \mathfrak{D}_\tau^S; \theta) \right] - \mathbb{E}_{q_2} \left[ \ell(\mathfrak{D}_\tau^Q, \mathfrak{D}_\tau^S; \theta) \right] \right| \tag{17a}$$

$$\leq \left| \int_{\tau \in \mathcal{T}_C} [q_1(\tau) - q_2(\tau)] \ell(\mathfrak{D}_\tau^Q, \mathfrak{D}_\tau^S; \theta) d\tau \right| \tag{17b}$$

$$+ \left| \int_{\tau \in \mathcal{T}_1} q_1(\tau) \ell(\mathfrak{D}_\tau^Q, \mathfrak{D}_\tau^S; \theta) d\tau - \int_{\tau \in \mathcal{T}_2} q_2(\tau) \ell(\mathfrak{D}_\tau^Q, \mathfrak{D}_\tau^S; \theta) d\tau \right| \tag{17c}$$

$$\leq \int_{\tau \in \mathcal{T}_C} \left| q_1(\tau) - q_2(\tau) \right| \ell(\mathfrak{D}_\tau^Q, \mathfrak{D}_\tau^S; \theta) d\tau \tag{17d}$$

$$+ \int_{\tau \in \mathcal{T}_1} \left| q_1(\tau) - q_2(\tau) \right| \ell(\mathfrak{D}_\tau^Q, \mathfrak{D}_\tau^S; \theta) d\tau + \int_{\tau \in \mathcal{T}_2} \left| q_1(\tau) - q_2(\tau) \right| \ell(\mathfrak{D}_\tau^Q, \mathfrak{D}_\tau^S; \theta) d\tau \tag{17e}$$

$$\leq 3\mathcal{L}_{\max} \int_{\tau \in \mathcal{T}} \left| q_1(\tau) - q_2(\tau) \right| d\tau = 3\mathcal{L}_{\max} d_{\mathcal{Q}_\alpha}(q_1, q_2), \tag{17f}$$

which implies $3\mathcal{L}_{\max}$-Lipschitz continuity of $g(q)$ w.r.t. $\forall q \in \mathcal{Q}_\alpha$.

According to the Remark in [77], there always exists the global Stackelberg equilibrium as the Definition 1 when $\mathcal{Q}_\alpha \times \Theta$ is compact and $\mathcal{F}(q, \theta)$ is continuous. This completes the proof of Proposition 2. ∎

## C.5  Proof of Quasi-concavity for $\mathcal{F}(q, \theta)$ *w.r.t.* $q$

It can be validated that $\mathcal{F}(q, \theta)$ is a quasi-concave function *w.r.t.* $q$, meaning that for any positive number $l \in \mathbb{R}_+$, the set $\{q | q \in \mathcal{Q}_\alpha, \mathcal{F}(q, \theta) > l\}$ is convex in $\mathcal{Q}_\alpha$.

***Proof:*** According to the conventional definition (i.e., the superlevel set is convex [78]), for all $\lambda_1 + \lambda_2 = 1$, $q_1, q_2 \in \{q | \mathcal{F}(q, \theta) > l\}$, we can have

$$\mathcal{F}(\lambda_1 q_1 + \lambda_2 q_2, \theta) = \mathbb{E}_{\lambda_1 q_1 + \lambda_2 q_2} \left[ \ell(\mathfrak{D}_\tau^Q, \mathfrak{D}_\tau^S; \theta) \right] \tag{18a}$$

$$= \lambda_1 \mathbb{E}_{q_1} \left[ \ell(\mathfrak{D}_\tau^Q, \mathfrak{D}_\tau^S; \theta) \right] + \lambda_2 \mathbb{E}_{q_2} \left[ \ell(\mathfrak{D}_\tau^Q, \mathfrak{D}_\tau^S; \theta) \right] \tag{18b}$$

$$= \lambda_1 \mathcal{F}(q_1, \theta) + \lambda_2 \mathcal{F}(q_2, \theta) \tag{18c}$$

$$> l. \tag{18d}$$

Thus, $\lambda_1 q_1 + \lambda_2 q_2 \in \{q | \mathcal{F}(q, \theta) > l\}$ and the superlevel set is convex, implying that $\mathcal{F}(q, \theta)$ is quasi-concave w.r.t. $q$. ∎

## C.6  Proof of Theorem 4.1

**Theorem 4.1** (Convergence Rate for the Second Player) *Let the iteration sequence in optimization be:* $\cdots \mapsto \{q_{t-1}, \theta_t\} \mapsto \{q_t, \theta_{t+1}\} \mapsto \cdots \mapsto \{q_*, \theta_*\}$, *with the converged equilibrium* $(q_*, \theta_*)$. *Under the Assumption 2 and suppose that* $\|I - \lambda \nabla_{\theta\theta}^2 \mathcal{F}(q_*, \theta_*)\|_2 <$

$1 - \lambda\beta_q\beta_h$, *we can have* $\lim_{t\to\infty} \frac{||\theta_{t+1}-\theta_*||_2}{||\theta_t-\theta_*||_2} \leq 1$, *and the iteration converges with the rate* $\left(||I - \lambda\nabla^2_{\theta\theta}\mathcal{F}(q_*,\theta_*)||_2 + \lambda\beta_q\beta_h\right)$.

***Proof:*** Let the resulting stationary point be $[q_*,\theta_*]$, we denote the difference terms by $\hat{q} = q - q_*$ and $\hat{\theta} = \theta - \theta_*$. Then, according to the optimization step, we can have the following equations:

$$\theta_{t+1} = \theta_t - \lambda\nabla_\theta\mathcal{F}(q_t;\theta_t) \implies \hat{\theta}_{t+1} = \hat{\theta}_t - \lambda\nabla_\theta\mathcal{F}(q_t;\theta_t). \tag{19}$$

Now we perform the first-order Taylor expansion of the $\theta$ related function $\nabla_\theta\mathcal{F}(q_t;\theta)$ around $\theta_*$ and can derive:

$$\nabla_\theta\mathcal{F}(q_t;\theta) = \nabla_\theta\mathcal{F}(q_t;\theta_*) + \nabla^2_{\theta\theta}\mathcal{F}(q_t;\theta_*)(\theta - \theta_*) + \mathcal{O}(||\theta - \theta_*||) \tag{20a}$$

$$\nabla_\theta\mathcal{F}(q_t;\theta_t) \simeq \nabla_\theta\mathcal{F}(q_t;\theta_*) + \nabla^2_{\theta\theta}\mathcal{F}(q_t;\theta_*)(\theta_t - \theta_*). \tag{20b}$$

Then we have the following result with the help of Assumption 2:

$$||\nabla_\theta\mathcal{F}(q_t;\theta_*)||_2 = ||\nabla_\theta\mathcal{F}(q_t;\theta_*) - \nabla_\theta\mathcal{F}(q_*;\theta_*)||_2 \tag{21a}$$

$$= ||\nabla_\theta\mathcal{F}(h(\theta_t);\theta_*) - \nabla_\theta\mathcal{F}(h(\theta_*);\theta_*)||_2 \tag{21b}$$

$$\leq \beta_q d_{\mathcal{Q}_\alpha}(h(\theta_t), h(\theta_*)) \tag{21c}$$

$$\leq \beta_q\beta_h||\theta_t - \theta_*||_2. \tag{21d}$$

With Eq. (19), Eq. (20) and Eq. (21), we can derive the equation that:

$$\hat{\theta}_{t+1} = \hat{\theta}_t - \lambda\nabla_\theta\mathcal{F}(q_t;\theta_t) \tag{22a}$$

$$= \hat{\theta}_t - \lambda\left[\nabla_\theta\mathcal{F}(q_t;\theta_*) + \nabla^2_{\theta\theta}\mathcal{F}(q_t;\theta_*)\hat{\theta}_t\right] \tag{22b}$$

$$= \left[I - \lambda\nabla^2_{\theta\theta}\mathcal{F}(q_t;\theta_*)\right]\hat{\theta}_t - \lambda\nabla_\theta\mathcal{F}(q_t;\theta_*) \tag{22c}$$

$$\implies ||\hat{\theta}_{t+1}||_2 \leq ||I - \lambda\nabla^2_{\theta\theta}\mathcal{F}(q_t;\theta_*)||_2||\hat{\theta}_t||_2 + \lambda||\nabla_\theta\mathcal{F}(q_t;\theta_*)||_2 \tag{22d}$$

$$\leq \left(||I - \lambda\nabla^2_{\theta\theta}\mathcal{F}(q_t;\theta_*)||_2 + \lambda\beta_q\beta_h\right)||\hat{\theta}_t||_2 \tag{22e}$$

Thus, when $||I - \lambda\nabla^2_{\theta\theta}\mathcal{F}(q_*;\theta_*)||_2 < 1 - \lambda\beta_q\beta_h$, we have

$$\lim_{t\to\infty}\frac{||\hat{\theta}_{t+1}||_2}{||\hat{\theta}_t||_2} \leq \lim_{t\to\infty}||I - \lambda\nabla^2_{\theta\theta}\mathcal{F}(q_t;\theta_*)||_2 + \lambda\beta_q\beta_h \tag{23a}$$

$$= ||I - \lambda\nabla^2_{\theta\theta}\mathcal{F}(q_*;\theta_*)||_2 + \lambda\beta_q\beta_h \tag{23b}$$

$$< 1. \tag{23c}$$

This completes the proof of Theorem 4.1. ∎

### C.7 Proof of Theorem 4.2

**Theorem 4.2** (Asymptotics in the Tail Risk Cases) *Under the Assumption 1 and given a batch of tasks* $\{\tau_i\}_{i=1}^{\mathcal{B}}$, *we can have*

$$CVaR_\alpha(\theta_T^{meta}) - CVaR_\alpha(\theta_*) \leq \beta_\tau||\theta_T^{meta} - \theta_*|| + \frac{VaR_\alpha^*}{1-\alpha}\Big(\mathbb{P}(\mathcal{T}_1) - \mathbb{P}(\mathcal{T}_2)\Big), \tag{24}$$

*where* $\mathcal{T}_1 = \{\tau : \ell^* < VaR_\alpha^*, \ell^{meta} \geq VaR_\alpha^{meta}\}, \mathcal{T}_2 = \{\tau : \ell^* \geq VaR_\alpha^*, \ell^{meta} < VaR_\alpha^{meta}\}$.

**Proof.** Given a batch of tasks $\{\tau_1, \cdots, \tau_{\mathcal{B}}\}$ and according to the definition of $\text{CVaR}_\alpha$, we have

$$\text{CVaR}_\alpha(\theta_T^{\text{meta}}) - \text{CVaR}_\alpha(\theta_*) \tag{25a}$$

$$= \frac{1}{1-\alpha} \int_{\{\tau:\ell^{\text{meta}} \geq \text{VaR}_\alpha^{\text{meta}}\}} \ell^{\text{meta}} p(\tau) d\tau - \frac{1}{1-\alpha} \int_{\{\tau:\ell^* \geq \text{VaR}_\alpha^*\}} \ell^* p(\tau) d\tau \tag{25b}$$

$$= \int_\tau \ell^{\text{meta}} p_\alpha(\tau; \theta_T^{\text{meta}}) d\tau - \int_\tau \ell^* p_\alpha(\tau; \theta_*) d\tau \tag{25c}$$

$$= \int_\tau \Big(\ell^{\text{meta}} p_\alpha(\tau; \theta_T^{\text{meta}}) - \ell^* p_\alpha(\tau; \theta_T^{\text{meta}})\Big) d\tau + \int_\tau \Big(\ell^* p_\alpha(\tau; \theta_T^{\text{meta}}) - \ell^* p_\alpha(\tau; \theta_*)\Big) d\tau \tag{25d}$$

$$= \int_\tau \Big(\ell^{\text{meta}} - \ell^*\Big) p_\alpha(\tau; \theta_T^{\text{meta}}) d\tau + \int_\tau \ell^* \Big(p_\alpha(\tau; \theta_T^{\text{meta}}) - p_\alpha(\tau; \theta_*)\Big) d\tau \tag{25e}$$

$$\leq \beta_\tau \|\theta_T^{\text{meta}} - \theta_*\| + \int_{\mathcal{T}_1 \cup \mathcal{T}_2 \cup \mathcal{T}_3 \cup \mathcal{T}_4} \ell^* \Big(p_\alpha(\tau; \theta_T^{\text{meta}}) - p_\alpha(\tau; \theta_*)\Big) d\tau \tag{25f}$$

$$= \beta_\tau \|\theta_T^{\text{meta}} - \theta_*\| + \int_{\mathcal{T}_1 \cup \mathcal{T}_2} \ell^* \Big(p_\alpha(\tau; \theta_T^{\text{meta}}) - p_\alpha(\tau; \theta_*)\Big) d\tau \tag{25g}$$

$$= \beta_\tau \|\theta_T^{\text{meta}} - \theta_*\| + \int_{\mathcal{T}_1} \ell^* p_\alpha(\tau; \theta_T^{\text{meta}}) d\tau - \int_{\mathcal{T}_2} \ell^* p_\alpha(\tau; \theta_*) d\tau \tag{25h}$$

$$= \beta_\tau \|\theta_T^{\text{meta}} - \theta_*\| + \frac{1}{1-\alpha} \int_{\mathcal{T}_1} \ell^* p(\tau) d\tau - \frac{1}{1-\alpha} \int_{\mathcal{T}_2} \ell^* p(\tau) d\tau \tag{25i}$$

$$\leq \beta_\tau \|\theta_T^{\text{meta}} - \theta_*\| + \frac{\text{VaR}_\alpha^*}{1-\alpha} \int_{\mathcal{T}_1} p(\tau) d\tau - \frac{\text{VaR}_\alpha^*}{1-\alpha} \int_{\mathcal{T}_2} p(\tau) d\tau \tag{25j}$$

$$= \beta_\tau \|\theta_T^{\text{meta}} - \theta_*\| + \frac{\text{VaR}_\alpha^*}{1-\alpha} \mathbb{P}(\mathcal{T}_1) - \frac{\text{VaR}_\alpha^*}{1-\alpha} \mathbb{P}(\mathcal{T}_2). \tag{25k}$$

In inequality (25f), $\mathcal{T}_1 = \{\tau : \ell^* < \text{VaR}_\alpha^*, \ell^{\text{meta}} \geq \text{VaR}_\alpha^{\text{meta}}\}, \mathcal{T}_2 = \{\tau : \ell^* \geq \text{VaR}_\alpha^*, \ell^{\text{meta}} < \text{VaR}_\alpha^{\text{meta}}\}, \mathcal{T}_3 = \{\tau : \ell^* < \text{VaR}_\alpha^*, \ell^{\text{meta}} < \text{VaR}_\alpha^{\text{meta}}\}, \mathcal{T}_4 = \{\tau : \ell^* \geq \text{VaR}_\alpha^*, \ell^{\text{meta}} \geq \text{VaR}_\alpha^{\text{meta}}\}$. Moreover, this inequality holds due to the $\beta_\tau$−Lipschitz continuous of $\ell(\mathfrak{D}_\tau; \theta)$. In Eq. (25g), $p_\alpha(\tau; \theta_T^{\text{meta}}) = p_\alpha(\tau; \theta_*) = 0$ when $\tau \in \mathcal{T}_3$, and $p_\alpha(\tau; \theta_T^{\text{meta}}) = p_\alpha(\tau; \theta_*) = \frac{p(\tau)}{1-\alpha}$ when $\tau \in \mathcal{T}_4$. Thus, we complete the proof of Theorem 4.2. ∎

### C.8 Proof of Theorem 4.3

**Theorem 4.3** (Generalization Bound in the Tail Risk Cases) *Given a collection of task samples $\{\tau_i\}_{i=1}^{\mathcal{B}}$ and corresponding meta datasets, we can derive the following generalization bound in the presence of tail risk:*

$$R(\theta_*) \leq \widehat{R}(\theta_*) + \sqrt{\frac{2\big(\frac{\alpha}{1-\alpha}\mathcal{L}_{\max}^2 + \mathbb{V}_{\tau_i \sim p_\alpha(\tau)}\big[\ell(\mathfrak{D}_{\tau_i}^Q, \mathfrak{D}_{\tau_i}^S; \theta_*)\big]\big) \ln\left(\frac{1}{\epsilon}\right)}{\mathcal{B}}} \\ + \frac{1}{3(1-\alpha)} \frac{\mathcal{L}_{\max}}{\mathcal{B}} \left(2\ln\left(\frac{1}{\epsilon}\right) + 3\alpha\mathcal{B}\right), \tag{26}$$

*where the inequality holds with probability at least $1 - \epsilon$ and $\epsilon \in (0, 1)$, $\mathbb{V}[\cdot]$ denotes the variance operation, and $\mathcal{L}_{\max}$ is from the Assumption 1.*

**Proof.** $R(\theta_*) - \widehat{R}(\theta_*)$ can be decomposed to two parts, i.e.,

$$R(\theta_*) - \widehat{R}(\theta_*) = \Big(R(\theta_*) - \widehat{R}_w(\theta_*)\Big) + \Big(\widehat{R}_w(\theta_*) - \widehat{R}(\theta_*)\Big). \tag{27}$$

For the first part (i.e., $R(\theta_*) - \widehat{R}_w(\theta_*)$), we will adopt the Bernstein's inequality to provide an upper bound. Regarding $\frac{p_\alpha(\tau_i)}{p(\tau_i)}\ell(\mathfrak{D}_{\tau_i}^Q, \mathfrak{D}_{\tau_i}^S; \theta_*) - R(\theta_*)$ as a random variable with respect to $\tau_i$ and according to Assumption 1, we know that

$$\frac{p_\alpha(\tau_i)}{p(\tau_i)}\ell(\mathfrak{D}_{\tau_i}^Q, \mathfrak{D}_{\tau_i}^S; \theta_*) - R(\theta_*) \leq \frac{1}{1-\alpha}\ell(\mathfrak{D}_{\tau_i}^Q, \mathfrak{D}_{\tau_i}^S; \theta_*) - R(\theta_*) \leq \frac{1}{1-\alpha}\mathcal{L}_{\max}. \tag{28}$$

Thus, following Bernstein's inequality [79], we know that

$$\mathbb{P}\left( \left| \frac{1}{\mathcal{B}} \sum_{i=1}^{\mathcal{B}} \frac{p_\alpha(\tau_i)}{p(\tau_i)} \ell(\mathfrak{D}_{\tau_i}^Q, \mathfrak{D}_{\tau_i}^S; \theta_*) - R(\theta_*) \right| \geq \xi \right) \tag{29a}$$

$$= \mathbb{P}\left( \left| \widehat{R}_w(\theta_*) - R(\theta_*) \right| \geq \xi \right) \tag{29b}$$

$$\leq \exp\left( -\frac{\mathcal{B}\xi^2}{2\mathbb{V}_{\tau_i}\left[ \frac{p_\alpha(\tau_i)}{p(\tau_i)} \ell(\mathfrak{D}_{\tau_i}^Q, \mathfrak{D}_{\tau_i}^S; \theta_*) - R(\theta_*) \right] + \frac{2}{3}\frac{1}{1-\alpha}\mathcal{L}_{\max}\xi} \right), \tag{29c}$$

where

$$\mathbb{V}_{\tau_i}\left[ \frac{p_\alpha(\tau_i)}{p(\tau_i)} \ell(\mathfrak{D}_{\tau_i}^Q, \mathfrak{D}_{\tau_i}^S; \theta_*) - R(\theta_*) \right] \tag{30a}$$

$$= \mathbb{V}_{\tau_i}\left[ \frac{p_\alpha(\tau_i)}{p(\tau_i)} \ell(\mathfrak{D}_{\tau_i}^Q, \mathfrak{D}_{\tau_i}^S; \theta_*) \right] \tag{30b}$$

$$= \mathbb{E}_{\tau_i}\left( \frac{p_\alpha(\tau_i)}{p(\tau_i)} \ell(\mathfrak{D}_{\tau_i}^Q, \mathfrak{D}_{\tau_i}^S; \theta_*) \right)^2 - \left( \mathbb{E}_{\tau_i}\left( \frac{p_\alpha(\tau_i)}{p(\tau_i)} \ell(\mathfrak{D}_{\tau_i}^Q, \mathfrak{D}_{\tau_i}^S; \theta_*) \right) \right)^2 \tag{30c}$$

$$= \int_\tau \left( \frac{p_\alpha(\tau)}{p(\tau)} \ell(\mathfrak{D}_\tau^Q, \mathfrak{D}_\tau^S; \theta_*) \right)^2 p(\tau)d\tau - \left( \int_\tau \frac{p_\alpha(\tau)}{p(\tau)} \ell(\mathfrak{D}_\tau^Q, \mathfrak{D}_\tau^S; \theta_*)p(\tau)d\tau \right)^2 \tag{30d}$$

$$= \int_\tau \frac{p_\alpha(\tau)}{p(\tau)} \ell(\mathfrak{D}_\tau^Q, \mathfrak{D}_\tau^S; \theta_*)^2 p_\alpha(\tau)d\tau - \left( \int_\tau \ell(\mathfrak{D}_\tau^Q, \mathfrak{D}_\tau^S; \theta_*)p_\alpha(\tau)d\tau \right)^2 \tag{30e}$$

$$= \frac{1}{1-\alpha} \int_\tau \ell(\mathfrak{D}_\tau^Q, \mathfrak{D}_\tau^S; \theta_*)^2 p_\alpha(\tau)d\tau - \left( \int_\tau \ell(\mathfrak{D}_\tau^Q, \mathfrak{D}_\tau^S; \theta_*)p_\alpha(\tau)d\tau \right)^2 \tag{30f}$$

$$\leq \left( \frac{1}{1-\alpha} - 1 \right) \int_\tau \ell(\mathfrak{D}_\tau^Q, \mathfrak{D}_\tau^S; \theta_*)^2 p_\alpha(\tau)d\tau + \mathbb{V}_{\tau \sim p_\alpha(\tau)}\left[ \ell(\mathfrak{D}_\tau^Q, \mathfrak{D}_\tau^S; \theta_*) \right] \tag{30g}$$

$$:= \frac{\alpha}{1-\alpha} \mathcal{L}_{\max}^2 + \mathbb{V}_{\tau \sim p_\alpha(\tau)}. \tag{30h}$$

Setting $\epsilon$ to match the upper bound in inequality (29c) shows that with probability at least $1 - \epsilon$, the following bound holds:

$$\left| \widehat{R}_w(\theta_*) - R(\theta_*) \right| \leq \sqrt{\frac{2(\frac{\alpha}{1-\alpha}\mathcal{L}_{\max}^2 + \mathbb{V}_{\tau \sim p_\alpha(\tau)})\ln\left(\frac{1}{\epsilon}\right)}{\mathcal{B}}} + \frac{2\mathcal{L}_{\max}\ln\left(\frac{1}{\epsilon}\right)}{3(1-\alpha)\mathcal{B}}. \tag{31}$$

For the second part (i.e., $\widehat{R}_w(\theta_*) - \widehat{R}(\theta_*)$), we have

$$\widehat{R}_w(\theta_*) - \widehat{R}(\theta_*) = \frac{1}{\mathcal{B}} \sum_{i=1}^{\mathcal{B}} \left( \frac{p_\alpha(\tau_i)}{p(\tau_i)} - \delta(\tau_i) \right) \ell(\mathfrak{D}_{\tau_i}^Q, \mathfrak{D}_{\tau_i}^S; \theta_*) \tag{32a}$$

$$\leq \frac{\mathcal{L}_{\max}}{\mathcal{B}} \sum_{i=1}^{\mathcal{B}} \left( \frac{p_\alpha(\tau_i)}{p(\tau_i)} - \delta(\tau_i) \right) \tag{32b}$$

$$= \frac{\mathcal{L}_{\max}}{\mathcal{B}} \sum_{i=1}^{\mathcal{B}} \left( \frac{p_\alpha(\tau_i)}{p(\tau_i)} - 1 \right)\delta(\tau_i) \tag{32c}$$

$$= \frac{\alpha}{1-\alpha} \frac{\mathcal{L}_{\max}}{\mathcal{B}} \sum_{i=1}^{\mathcal{B}} \delta(\tau_i). \tag{32d}$$

In summary, we can obtain an upper bound of $R(\theta_*) - \widehat{R}(\theta_*)$. That is,

$$
\begin{aligned}
R(\theta_*) - \widehat{R}(\theta_*) &\leq \left|\widehat{R}_w(\theta_*) - R(\theta_*)\right| + \widehat{R}_w(\theta_*) - \widehat{R}(\theta_*) \\
&\leq \sqrt{\frac{2(\frac{\alpha}{1-\alpha}\mathcal{L}_{\max}^2 + \mathbb{V}_{\tau \sim p_\alpha(\tau)})\ln\left(\frac{1}{\epsilon}\right)}{\mathcal{B}}} + \frac{2\mathcal{L}_{\max}\ln\left(\frac{1}{\epsilon}\right)}{3(1-\alpha)\mathcal{B}} + \frac{\alpha}{1-\alpha}\frac{\mathcal{L}_{\max}}{\mathcal{B}}\sum_{i=1}^{\mathcal{B}}\delta(\tau_i) \\
&\leq \sqrt{\frac{2(\frac{\alpha}{1-\alpha}\mathcal{L}_{\max}^2 + \mathbb{V}_{\tau \sim p_\alpha(\tau)})\ln\left(\frac{1}{\epsilon}\right)}{\mathcal{B}}} + \frac{1}{3(1-\alpha)}\frac{\mathcal{L}_{\max}}{\mathcal{B}}\left(2\ln\left(\frac{1}{\epsilon}\right) + 3\alpha\mathcal{B}\right).
\end{aligned}
$$

This completes the proof of Theorem 4.3. ∎

## C.9 Proof of Theorem 4.4

**Theorem 4.4** *Let $F_{\ell\text{-}KDE}^{-1}(\alpha;\theta) = VaR_\alpha^{KDE}[\ell(\mathcal{T},\theta)]$ and $F_\ell^{-1}(\alpha;\theta) = VaR_\alpha[\ell(\mathcal{T},\theta)]$. Suppose that $K(x)$ is lower bounded by a constant, $\forall x$. For any $\epsilon > 0$, with probability at least $1 - \epsilon$, we can have the following bound:*

$$
\sup_{\theta \in \Theta}\left(F_{\ell\text{-}KDE}^{-1}(\alpha;\theta) - F_\ell^{-1}(\alpha;\theta)\right) \leq \mathcal{O}\left(\frac{h_\ell}{\sqrt{\mathcal{B} * \log \mathcal{B}}}\right). \tag{33}
$$

*Proof.* For any constant $M$, we firstly notice that

$$
\mathbb{P}_{\tau_1,\cdots,\tau_\mathcal{B}}\left(\sup_{\theta \in \Theta}\left(F_{\ell\text{-}KDE}^{-1}(\alpha;\theta) - F_\ell^{-1}(\alpha;\theta)\right) \leq M\right) \geq 1 - \epsilon \tag{34a}
$$

$$
\Leftrightarrow \mathbb{P}_{\tau_1,\cdots,\tau_\mathcal{B}}\left(\sup_{\theta \in \Theta}\left(F_{\ell\text{-}KDE}^{-1}(\alpha;\theta) - F_\ell^{-1}(\alpha;\theta)\right) \geq M\right) \leq \epsilon \tag{34b}
$$

$$
\Leftrightarrow \mathbb{P}_{\tau_1,\cdots,\tau_\mathcal{B}}\left(\sup_{\theta \in \Theta}\left(F_\ell(t - M;\theta) - F_{\ell\text{-}KDE}(t;\theta)\right) \geq 0\right) \leq \epsilon, \quad t = F_{\ell\text{-}KDE}^{-1}(\alpha;\theta). \tag{34c}
$$

For any $\theta$ and $t$, we have

$$
\mathbb{P}_{\tau_1,\cdots,\tau_\mathcal{B}}\left(F_\ell(t - M;\theta) - F_{\ell\text{-}KDE}(t;\theta) \geq 0\right) \leq \epsilon \tag{35a}
$$

$$
\Leftrightarrow \mathbb{P}_{\tau_1,\cdots,\tau_\mathcal{B}}\left(F_\ell(t - M;\theta) - F_{\ell\text{-}KDE}(t - M;\theta) + F_{\ell\text{-}KDE}(t - M;\theta) - F_{\ell\text{-}KDE}(t;\theta) \geq 0\right) \leq \epsilon \tag{35b}
$$

$$
\Leftrightarrow \mathbb{P}_{\tau_1,\cdots,\tau_\mathcal{B}}\left(F_\ell(t - M;\theta) - F_{\ell\text{-}KDE}(t - M;\theta) \geq F_{\ell\text{-}KDE}(t;\theta) - F_{\ell\text{-}KDE}(t - M;\theta)\right) \leq \epsilon \tag{35c}
$$

$$
\Leftrightarrow \mathbb{P}_{\tau_1,\cdots,\tau_\mathcal{B}}\left(F_\ell(t;\theta) - F_{\ell\text{-}KDE}(t;\theta) \geq F_{\ell\text{-}KDE}(t + M;\theta) - F_{\ell\text{-}KDE}(t;\theta)\right) \leq \epsilon \tag{35d}
$$

$$
\Leftrightarrow \mathbb{P}_{\tau_1,\cdots,\tau_\mathcal{B}}\left(F_\ell(t;\theta) - F_{\ell\text{-}KDE}(t;\theta) \geq \frac{M}{\mathcal{B}h_\ell}\left(\sum_{i=1}^{\mathcal{B}}K\left(\frac{t - \ell(\mathfrak{D}_{\tau_i}^Q, \mathfrak{D}_{\tau_i}^S;\theta)}{h_\ell}\right)\right) + o(M)\right) \leq \epsilon \tag{35e}
$$

$$
\Leftarrow \mathbb{P}_{\tau_1,\cdots,\tau_\mathcal{B}}\left(F_\ell(t;\theta) - F_{\ell\text{-}KDE}(t;\theta) \geq \frac{M\mathcal{K}_{\min}}{h_\ell}\right) \leq \epsilon, \tag{35f}
$$

where $\mathcal{K}_{\min}$ is the lower bound of the kernel function $K(x)$, *i.e.*, $K(x) \geq \mathcal{K}_{\min}, \forall x$.

According to Theorem 3 of [80], we know that for any $\epsilon > 0$, with probability at least $1 - \epsilon$, the following inequality holds:

$$
\mathbb{P}_{\tau_1,\cdots,\tau_\mathcal{B}}\left(\sup_{\theta \in \Theta, t \geq 0}\left(F_\ell(t;\theta) - F_{\ell\text{-}KDE}(t;\theta)\right) \geq \frac{C}{\sqrt{\mathcal{B} * \log \mathcal{B}}}\right) \leq \epsilon, \tag{36}
$$

where $C$ is a constant. Let $M = \frac{h_\ell C}{\mathcal{K}_{\min}\sqrt{\mathcal{B} * \log \mathcal{B}}}$. Thus, the Eq. (35f) holds and we complete the proof of Theorem 4.4. ∎

## C.10 Additional Theorem

To gain more theoretical insights into a popular meta-learning method—MAML [25], we provide the following Theorem C.1. Before proceeding, we introduce some notations. During meta-training, a

finite number of task instances are observed by first sampling a task from the distribution $p(\tau)$. Each task $D_{\tau_i}$ comprises a collection of $m_i$ data points $\{(\mathfrak{D}_{i,j}^S, \mathfrak{D}_{i,j}^Q)\}_{j=1}^{m_i}$, which are distributed over $\mathcal{Z}$ with each data point drawn from a distribution $D_i$. For some risk $\ell$, define the family of functions $\mathcal{F}_{\mathcal{Z}} := \{\ell(\theta - \lambda\nabla_\theta\ell(\mathfrak{D}_{\tau_i}^S; \theta), \mathfrak{D}_{\tau_i}^Q) : \theta \in \Theta\}$. For each task $D_{\tau_i}$, the Rademacher complexity of $\mathcal{F}$ on $m_i$ samples is

$$\mathcal{R}_{m_i}^i(\mathcal{F}_{\mathcal{Z}}) = \mathbb{E}_{(\mathfrak{D}_{i,j}^S, \mathfrak{D}_{i,j}^Q)\sim(D_i)^{m_i}}\mathbb{E}_{\boldsymbol{\epsilon}}\left[\sup_{\theta\in\Theta}\frac{1}{m_i}\sum_{j=1}^{m_i}\epsilon_j\ell(\theta - \lambda\nabla_\theta\ell(\mathfrak{D}_{i,j}^S; \theta), \mathfrak{D}_{i,j}^Q)\right], \quad (37)$$

where the $\epsilon_j$'s are Rademacher random variables. Let $F_i(\theta) = \mathbb{E}_{D_i}\ell(\theta - \lambda\nabla_\theta\ell(\mathfrak{D}_{i,j}^S; \theta), \mathfrak{D}_{i,j}^Q)$, $\hat{F}_i(\theta) = \frac{1}{m_i}\sum_{j=1}^{m_i}\ell(\theta - \lambda\nabla_\theta\ell(\mathfrak{D}_{i,j}^S; \theta), \mathfrak{D}_{i,j}^Q)$. Denote by $\theta^*$ the optimal model parameter under the two-stage algorithm. Theorem C.1 provides generalization of the algorithm to new tasks.

**Theorem C.1** (Generalization Bound for MAML in the Tail Risk Cases). *For a new task $\tau_{\mathcal{B}+1}$ with distribution $D_{\mathcal{B}+1}$, if $D_{\mathcal{B}+1} = \sum_{i=1}^{\mathcal{B}} a_i D_i$, then with probability at least $1 - \delta$ for any $\delta > 0$, we can have*

$$F_{\mathcal{B}+1}(\theta^*) \leq \max_i \hat{F}_i(\theta^*) + \sum_{i=1}^{\mathcal{B}}\left[2a_i\mathcal{R}_{m_i}^i(\mathcal{F}_{\mathcal{Z}}) + a_i\sqrt{\frac{\log(\mathcal{B}/\delta)}{2m_i}}\right]. \quad (38)$$

***Proof.*** The proof consists of two parts. We first explore the generalization to new instances of previously-seen tasks. Then we solve the generalization to new tasks.

**Step 1.** For any sample set $\mathcal{A} = \{(\mathfrak{D}_{i,j}^S, \mathfrak{D}_{i,j}^Q)\}_{j=1}^{m_i}$, define $\Phi(\mathcal{A}) = \sup_{\ell\in\mathcal{F}_{\mathcal{Z}}} F_i(\theta) - \hat{F}_i(\theta)$. Let $\mathcal{A}$ and $\mathcal{A}' := \{((\mathfrak{D}_{i,j}^S)', (\mathfrak{D}_{i,j}^Q)')\}_{j=1}^{m_i}$ be two samples that differ by exactly one point. According to the fact $\sup_x f(x) - \sup_x g(x) \leq \sup_x(f(x) - g(x))$, we know that $\Phi(\mathcal{A}') - \Phi(\mathcal{A}) \leq \frac{1}{m_i}$ due to the difference in exactly one point. Similarly, we can obtain $\Phi(\mathcal{A}) - \Phi(\mathcal{A}') \leq \frac{1}{m}$, thus $|\Phi(\mathcal{A}) - \Phi(\mathcal{A}')| \leq \frac{1}{m}$. Following McDiarmid's inequality, for any $\delta > 0$, with probability at least $1 - \frac{\delta}{2}$, we have

$$\Phi(\mathcal{A}) \leq \mathbb{E}_{\mathcal{A}}[\Phi(\mathcal{A})] + \sqrt{\frac{\log(2/\delta)}{2m_i}}. \quad (39)$$

We next bound the expectation of the right-hand side of inequality (39) as follows:

$$\mathbb{E}_{\mathcal{A}}[\Phi(\mathcal{A})] = \mathbb{E}_{\mathcal{A}}\left[\sup_{\ell\in\mathcal{F}_{\mathcal{Z}}} F_i(\theta) - \hat{F}_i(\theta)\right]$$

$$= \mathbb{E}_{\mathcal{A}}\left[\sup_{\ell\in\mathcal{F}_{\mathcal{Z}}}\mathbb{E}_{\mathcal{A}'}\left[\mathbb{E}_{\mathcal{A}'}(F_i(\theta)) - \hat{F}_i(\theta)\right]\right] \quad (40)$$

$$\leq \mathbb{E}_{\mathcal{A},\mathcal{A}'}\left[\sup_{\ell\in\mathcal{F}_{\mathcal{Z}}}\mathbb{E}_{\mathcal{A}'}(F_i(\theta)) - \hat{F}_i(\theta)\right] \quad (41)$$

$$= \mathbb{E}_{\mathcal{A},\mathcal{A}'}\left[\sup_{\ell\in\mathcal{F}_{\mathcal{Z}}}\frac{1}{m_i}\sum_{j=1}^{m_i}\left(\ell(\mathfrak{D}_{i,j}^S, \mathfrak{D}_{i,j}^Q) - \ell((\mathfrak{D}_{i,j}^S)', (\mathfrak{D}_{i,j}^Q)')\right)\right] \quad (42)$$

$$= \mathbb{E}_{\mathcal{A},\mathcal{A}',\boldsymbol{\epsilon}}\left[\sup_{\ell\in\mathcal{F}_{\mathcal{Z}}}\frac{1}{m_i}\sum_{j=1}^{m_i}\epsilon_j\left(\ell(\mathfrak{D}_{i,j}^S, \mathfrak{D}_{i,j}^Q) - \ell((\mathfrak{D}_{i,j}^S)', (\mathfrak{D}_{i,j}^Q)')\right)\right]$$

$$\leq \mathbb{E}_{\mathcal{A},\boldsymbol{\epsilon}}\left[\sup_{\ell\in\mathcal{F}_{\mathcal{Z}}}\frac{1}{m_i}\sum_{j=1}^{m_i}\epsilon_j\ell(\mathfrak{D}_{i,j}^S, \mathfrak{D}_{i,j}^Q)\right] + \mathbb{E}_{\mathcal{A}',\boldsymbol{\epsilon}}\left[\sup_{\ell\in\mathcal{F}_{\mathcal{Z}}}\frac{1}{m_i}\sum_{j=1}^{m_i}\epsilon_j\ell((\mathfrak{D}_{i,j}^S)', (\mathfrak{D}_{i,j}^Q)')\right]$$

$$= 2\mathbb{E}_{\mathcal{A},\boldsymbol{\epsilon}}\left[\sup_{\ell\in\mathcal{F}_{\mathcal{Z}}}\frac{1}{m_i}\sum_{j=1}^{m_i}\epsilon_j\ell(\mathfrak{D}_{i,j}^S, \mathfrak{D}_{i,j}^Q)\right]$$

$$= 2\mathcal{R}_{m_i}(\mathcal{F}_{\mathcal{Z}}).$$

Eq. (40) uses the *law of total expectation*. Inequality (41) holds by Jensen's inequality and the convexity of the supremum function. In Eq. (42), $\ell(\mathfrak{D}_{i,j}^S, \mathfrak{D}_{i,j}^Q) := \ell(\theta - \lambda\nabla_\theta\ell(\mathfrak{D}_{i,j}^S; \theta), \mathfrak{D}_{i,j}^Q)$,

where $(\mathfrak{D}_{i,j}^S, \mathfrak{D}_{i,j}^Q) \in \mathcal{A}$. Following inequality (39), we can know that

$$F_i(\theta) \leq \hat{F}_i(\theta) + 2\mathcal{R}_{m_i}(\mathcal{F}_{\mathcal{Z}}) + \sqrt{\frac{\log(2/\delta)}{2m_i}}. \tag{43}$$

**Step 2.** Since the new distribution $D_{\mathcal{B}+1}$ is the convex combination of $D_i, \forall i = 1, \cdots, \mathcal{B}$, we have $F_{\mathcal{B}+1}(\theta) = \sum_{i=1}^{\mathcal{B}} a_i F_i(\theta)$. Accordingly, with probability at least $1 - \delta$ over the choice of samples used to compute $\hat{F}(\theta)$,

$$F_{\mathcal{B}+1}(\theta^*) = \sum_{i=1}^{\mathcal{B}} a_i F_i(\theta^*) \leq \sum_{i=1}^{\mathcal{B}} \left[ a_i \hat{F}_i(\theta^*) + 2a_i \mathcal{R}_{m_i}(\mathcal{F}_{\mathcal{Z}}) + a_i \sqrt{\frac{\log(2/\delta)}{2m_i}} \right], \tag{44}$$

which yields that

$$F_{\mathcal{B}+1}(\theta^*) \leq \max_i \hat{F}_i(\theta^*) + \sum_{i=1}^{\mathcal{B}} \left[ 2a_i \mathcal{R}_{m_i}^i(\mathcal{F}_{\mathcal{Z}}) + a_i \sqrt{\frac{\log(2/\delta)}{2m_i}} \right]. \tag{45}$$

In summary, the two steps complete the proof of Theorem C.1. ∎

## D   Implementation Details

### D.1   Benchmark Details & Neural Architectures & Opensource Codes

Here, we illustrate all meta learning benchmark purposes in Fig. 10, which includes sinusoid regression, pendulum system identification, few-shot image classification, and meta reinforcement learning. We no longer run experiments on the Omniglot dataset, as most baselines can achieve SOTA performance and cannot tell the difference well from the openreview of [1].

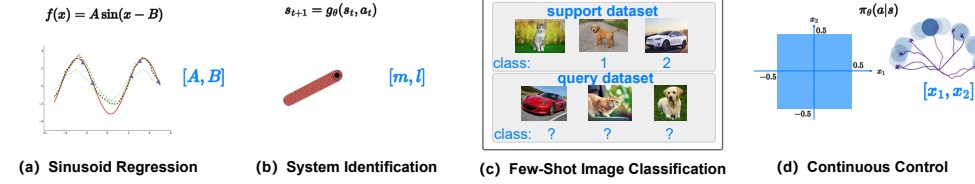

Figure 10: **Typical meta learning benchmarks in evaluation.**

**Sinusoid regression:** In [1, 42], a lot of easy tasks and limited challenging tasks are sampled for meta-training, with the tasks from the whole space employed in evaluation. The default range of the phase parameter is $B \in [0, \pi]$, while those of the amplitude are $A \in [0.1, 1.05]$ for easy tasks and $A \in [4.95, 5.0]$ for challenging tasks. Generally, sinusoid functions with larger amplitudes are hard to adapt from a few support data points. The mean square error works as the risk function to measure the gap between the predicted value $f(x)$ and the actual value. We set the task batch 50 for 5-shot and 25 for 10-shot, and the maximum iteration number is 70000. We refer the reader to TR-MAML and DR-MAML for all of the setups.

We retain the neural architectures [42, 1] in for all MAML like methods. In detail, all methods take a multilayer perceptron with two hidden layers and 40 ReLU activation units in each layer. The inner loop is achieved via one stochastic gradient descent step. As for CNP like methods, please refer to the vanilla set-up in [15] (The Github link is attached here: https://github.com/google-deepmind/neural-processes).

As the task space is hugh, there is no way to exactly estimate the risk quantile. Hence, the Oracle quantile in Fig. 5 is roughly computed from the sampled 100 tasks given the pretrained DR-MAML+. The rationale behind this operation is that increasing the population number in statistics reduces the quantile estimate bias.

**System identification:** The pendulum system is a classical environment in the OpenAI gym (environment details are: https://github.com/openai/gym/blob/master/gym/envs/classic_

`control/pendulum.py`), and it is an actuated joint with one fixed end. The goal of system identification for the pendulum system is to predict the state transition given arbitrary actions with several randomly collected transitions as the support dataset. The observation is a tuple in the form $(\cos\theta, \sin\theta, \theta')$, where $\theta \in [\pi, \pi]$. The action is in the range $a \in [-2.0, 2.0]$ and the torque is applied to the pendulum body. The mass $m$ and the length $l$ of the pendulum follows a uniform distribution $(m, l) \sim \mathcal{U}([0.4, 1.6] \times [0.4, 1.6])$, sampled variables configure a Markov decision process as the task. In each batch, there are 16 tasks, and each task comprises 200 data points. Specifically, 10 few-shot data points are randomly sampled to enable system identification per task, denoted as `10-shot`. For `20-shot` cases, the number of data points in support dataset is 20. And the maximum iteration number is 5000.

For all MAML-like methods, the neural architecture used here is a multilayer perceptron with three hidden layers of 128 hidden units each and the activation function is ReLU. The learning rate for both the inner and outer loops is set at 1e-4.

**Few-shot image classification:** The few-shot image classification is mostly described as an `N-way` `K-shot` classification, where `N` classes with `K`-labeled instances for each are considered. The dataset is organized in the same manner as that in [81, 42, 1]: These include 64 classes for meta-training, with the rest 36 classes for meta-testing. We generate each task in the way: 8 meta-training tasks from the class $\{6, 7, 7, 8, 8, 9, 9, 10\}$ are randomly generated from 64 meta-training classes; the remaining classes are organized similarly. As a result, each task is constructed from sampling one image from five classes, corresponding to a `5-way 1-shot` problem. The task batch is set 4 with a maximum number of iterations of 60000 in meta-training.

For all MAML-like methods, the neural architecture used here is a four-layer convolutional neural network for the *mini*-ImageNet datasets. The inner loop is achieved via one stochastic gradient descent step. We refer the reader to TR-MAML and DR-MAML for all of the setups (The Github link is attached here `https://github.com/lgcollins/tr-maml`).

**Meta reinforcement learning:** 2D Navigation is a classical meta reinforcement learning benchmark where efficient explorations matter. The task in 2D Navigation is to guide the point robot to take move actions for a purpose of reaching a specific goal location from the step-wise reward. The reward the agent receives from the environment is based on the distance to the goal, and 20 episodes work as the support dataset for navigation fast adaptation. In terms of the task distribution, we sample tasks from a uniform distribution $\mathcal{U}([-0.5, 0.5] \times [-0.5, 0.5])$ over goal locations.

As for the neural architecture for policy network set-ups, we refer the reader to vanilla MAML (Github link is attached here `https://github.com/tristandeleu/pytorch-maml-rl`) and CAVIA (The Github link is attached here `https://github.com/lmzintgraf/cavia/tree/master/rl`). And trust region policy optimization works for policy optimization.

Table 4: **Computational and memory cost in MaPLe relevant experiments.**

| Method | MaPLe | DR-MaPLe | DR-MaPLe+ (Ours) |
|---|---|---|---|
| Implementation Time | 2.1 h | +1.7 h | +1.7 h |
| Memory Usage | 41.57 G | +36.84G | +36.84G |

**Few-Shot Image Classification with MaPLe [69]:** The stochastic gradient descent is the default optimizer with the learning rate 0.0035, and A6000 GPUs work for computations. We examine tail task risk minimization effectiveness on three large datasets. The class number split setup in datasets (class number to train/validate/test) is TieredImageNet (351/97/160) [82], ImagenetA (128/32/40) [83], and ImagenetSketch (640/160/200) [84]. Table 4 reports the overall training time and memory, where the vanilla MaPLe serves as the anchor point, and + means additional costs from the two-stage operation. For details of experimental implementations and setups, feel free to access our code at `https://github.com/lvyiqin/DRMAML`.

### D.2 Modules in Python

This subsection includes the impelemention of KDE for the studied strategy. Here, the example of the hinge loss is illustrated as follows.

```
1  import numpy as np
2  import torch
3  from scipy.stats import gaussian_kde
4  from scipy.optimize import brentq
5
6  def loss(batch_loss, confidence_level):
7      # estimate the VaR_alpha according to kernel density estimator
8      kde = gaussian_kde(batch_loss)
9      try:
10         target_func = lambda x: kde.integrate_box_1d(-np.inf, x) -
   confidence_level
11         VaR_alpha = brentq(target_func, np.min(batch_loss), np.max(
   batch_loss))
12     except ValueError:
13         x = np.linspace(np.min(batch_loss), np.max(batch_loss), 1000)
14         pdf = kde.evaluate(x)
15         cdf = np.cumsum(pdf) / np.sum(pdf)
16         index = np.argmax(cdf >= confidence_level)
17         VaR_alpha = x[index]
18
19     # calculate the meta loss
20     tail_loss = [i - VaR_alpha if (i - VaR_alpha) > 0 else torch.
   tensor(0.).cuda() for i in batch_loss]
21     new_batch_loss = torch.stack(tail_loss).mean()
22     factor = 1 / (1 - confidence_level)
23     loss_meta = VaR_alpha + factor * new_batch_loss
24     return loss_meta
```

Listing 1: The calculation process of CVaR$_\alpha$ objective.

# E   Additional Experimental Results

Due to the page limit in the main paper, we include additional experiments and corresponding results in this section.

## E.1   Evaluation with Other Robust Meta Learners

In addition to MAML, we apply a similar modification to CNP, which results in TR-CNP, DRO-CNP, DR-CNP, and DR-CNP+ (DR-CNP with KDE for VaR$_\alpha$ estimates). We report the meta testing results on sinusoid regression and pendulum system identification benchmarks.

As illustrated in Table 5/6, all methods achieve comparable average performance in sinusoid and pendulum system identification. Regarding CVaR$_\alpha$, DR-CNP's improvement is relatively marginal over others except DR-CNP+. Compared to MAML, CNP seems more sensitive to quantile estimate accuracies when meeting with the studied strategies.

Table 5: **MSEs for Sinusoid** `5-shot` **with reported standard deviations (5 runs).** With $\alpha = 0.7$, the best results are in bold.

| Method | Average | Worst | CVaR$_\alpha$ |
|---|---|---|---|
| CNP [15] | $0.09_{\pm 0.00}$ | $2.71_{\pm 0.54}$ | $0.24_{\pm 0.01}$ |
| TR-CNP [42] | $0.10_{\pm 0.01}$ | $1.51_{\pm 0.30}$ | $0.22_{\pm 0.03}$ |
| DRO-CNP [66] | $0.09_{\pm 0.02}$ | $2.54_{\pm 1.81}$ | $0.21_{\pm 0.05}$ |
| DR-CNP [1] | $0.09_{\pm 0.01}$ | $1.62_{\pm 0.45}$ | $0.20_{\pm 0.02}$ |
| DR-CNP+(Ours) | $\mathbf{0.08}_{\pm \mathbf{0.01}}$ | $\mathbf{1.47}_{\pm \mathbf{0.90}}$ | $\mathbf{0.17}_{\pm \mathbf{0.02}}$ |

## E.2   Numeric Results in Tables and Histograms

As the improving tricks in this work are regarding the quantile estimators, here we particularly include the quantitative results to show the difference between DR-MAML and DR-MAML+ in Table 7/8.

Table 6: **MSEs for Pendulum** `10-shot` **with reported standard deviations (5 runs).** With $\alpha = 0.5$, the best results are in bold.

| Method | Average | Worst | CVaR$_\alpha$ |
|---|---|---|---|
| CNP [15] | 0.75$\pm$0.01 | 1.51$\pm$0.23 | 0.87$\pm$0.02 |
| TR-CNP [42] | 0.76$\pm$0.00 | **1.24$\pm$0.02** | 0.85$\pm$0.01 |
| DRO-CNP [66] | 0.73$\pm$0.01 | 1.51$\pm$0.16 | 0.85$\pm$0.01 |
| DR-CNP [1] | 0.75$\pm$0.01 | 1.40$\pm$0.16 | 0.86$\pm$0.01 |
| DR-CNP+(Ours) | **0.72$\pm$0.01** | 1.36$\pm$0.07 | **0.82$\pm$0.00** |

Note that the studied distributionally robust strategy is on the tail risk minimization, and CVaR$_\alpha$ is the direct optimization indicator. As can be seen, DR-MAML+'s performance superiority over DR-MAML is significant *w.r.t.* CVaR$_\alpha$ values in `5-shot` sinusoid regression and four *mini*-ImageNet meta-testing tasks. These scenarios are more challenging than others as (i) the context information for adaptation is limited in `5-shot` data points and (ii) the distributional shift is severe in *mini*-ImageNet meta-testing phase.

Table 7: **Test average mean square errors (MSEs) with reported standard deviations for sinusoid regression (5 runs).** We respectively consider `5-shot` and `10-shot` cases with $\alpha = 0.7$. The results are evaluated across the 490 meta-test tasks, as in [42]. The best results are in bold.

| Method | 5-shot | | | 10-shot | | |
|---|---|---|---|---|---|---|
| | Average | Worst | CVaR$_\alpha$ | Average | Worst | CVaR$_\alpha$ |
| DR-MAML [1] | 0.89$\pm$0.04 | 2.91$\pm$0.46 | 1.76$\pm$0.02 | **0.54$\pm$0.01** | 1.70$\pm$0.17 | 0.96$\pm$0.01 |
| DR-MAML+(Ours) | **0.87$\pm$0.02** | **2.78$\pm$0.22** | **1.65$\pm$0.02** | 0.59$\pm$0.02 | **1.51$\pm$0.11** | **0.95$\pm$0.02** |

Table 8: **Average** `5-way 1-shot` **classification accuracies in** *mini*-**ImageNet with reported standard deviations (3 runs).** With $\alpha = 0.5$, the best results are in bold. The higher, the better for all values.

| Method | Eight Meta-Training Tasks | | | Four Meta-Testing Tasks | | |
|---|---|---|---|---|---|---|
| | Average | Worst | CVaR$_\alpha$ | Average | Worst | CVaR$_\alpha$ |
| DR-MAML [1] | 70.2$\pm$0.2 | 63.4$\pm$0.2 | 67.2$\pm$0.1 | 49.4$\pm$0.1 | 47.1$\pm$0.1 | 47.5$\pm$0.1 |
| DR-MAML+(Ours) | **70.4$\pm$0.1** | **63.8$\pm$0.2** | **67.5$\pm$0.1** | **49.9$\pm$0.1** | **47.2$\pm$0.1** | **48.1$\pm$0.1** |

We can attribute the performance differences of the two methods to the cumulative quantile estimation errors using the crude MC. Even though the quantile estimation error in Fig. 5 difference is tiny in each step, the cumulative error indeed affects the converged equilibrium a lot. This reflects the advantage of the KDE's used in DR-MAML+ when the task batch size cannot be set larger in practice.

We also investigate the task risk value distributions in pendulum system identification. To this end, we visualize one run testing results for all methods in Fig. 11. It seems DR-MAML+'s result is more skewed to the left than others.

Fig. 12 displays all methods' performance *w.r.t.* the average and CVaR$_\alpha$ returns along the meta-training process. We exclude TR-MAML in visualization due to its worse performance and unstable training properties. We can find that the DR-MAML exhibits a fast performance rise at the early stage but its capability to continuously improve diminishes over time. DR-MAML+ consistently outperforms other baselines in most cases. The above suggests that the KDE module achieves performance gains over the crude MC when implemented with the two-stage distributionally robust strategy for meta RL scenarios.

## E.3 Sensitivity Analysis to Confidence Level

To reveal the impact of confidence levels on model performance, we perform a sensitivity analysis with respect to confidence levels. Since only DR-MAML and DR-MAML+ are influenced by the confidence levels during the distributionally robust optimization across all baselines, we only compare the performance of the two methods to highlight the differences between them. As shown in Fig.

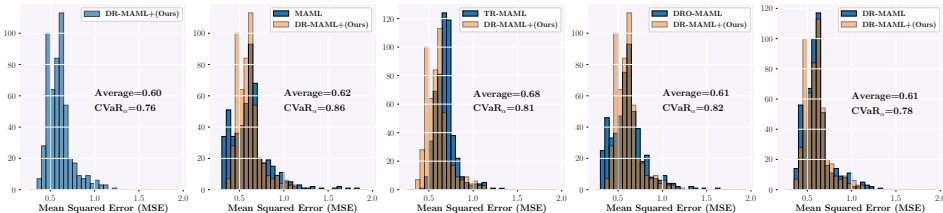

Figure 11: **Histograms of meta-testing performance in system identification.** With $\alpha = 0.5$, we visualize the comprision results of baselines and our DR-MAML+ in 10-shot prediction. The lower, the better for Average and $\text{CVaR}_\alpha$ values.

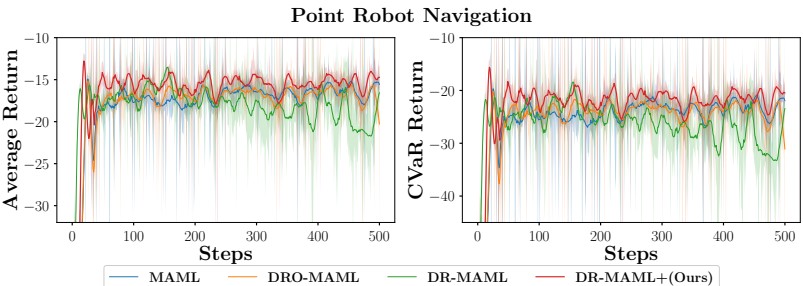

Figure 12: **Learning curves for the point robot navigation task.** Here, 20 trajectories work as the support set for adaptation. The curves report the normalized returns and are averaged over four random seeds, with $\alpha = 0.5$.

, we can observe that in both sinusoid 5-shot and 10-shot tasks, as the confidence level varies, DR-MAML+ exhibits more stable performance than DR-MAML, indicating that DR-MAML+ has a lower sensitivity to confidence levels. It can be illustrated that the crude Monte Carlo used in DR-MAML is more unstable in terms of quantile estimation than the kernel density estimator used in DR-MAML+. This can be due to the fact that the crude Monte Carlo method is more likely to get stuck in the local optimal solution. In addition, it can be seen from Fig.  that the performance of our developed DR-MAML+ is better than DR-MAML in most cases. DR-MAML+ exhibits lower mean squared errors than DR-MAML in the average, worst, and $\text{CVaR}_\alpha$ indicators, demonstrating the advantages of more accurate quantile estimation in improving robustness.

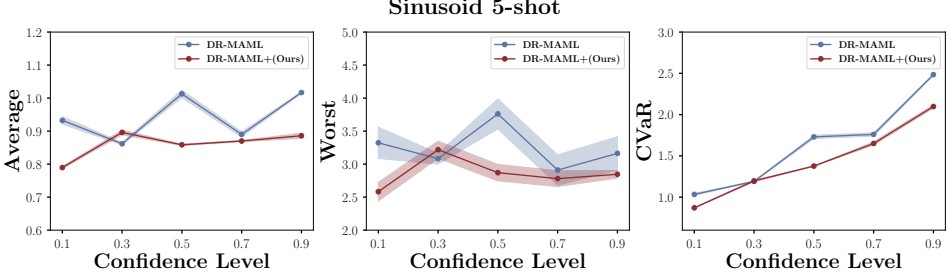

Figure 13: **Meta testing performance of DR-MAML and DR-MAML+ with different confidence level on Sinusoid 5-shot tasks.** In the plots, the vertical axis is the MSEs, the horizontal axis is the confidence level, and the shaded area represents the standard deviation.

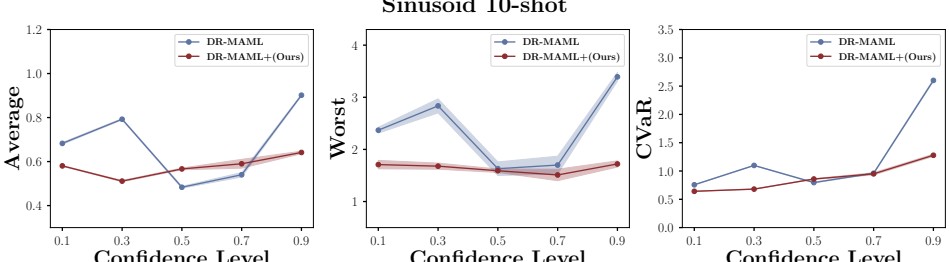

Figure 14: **Meta testing performance of DR-MAML and DR-MAML+ with different confidence level on Sinusoid 10-shot tasks.** In the plots, the vertical axis is the MSEs, the horizontal axis is the confidence level, and the shaded area represents the standard deviation.

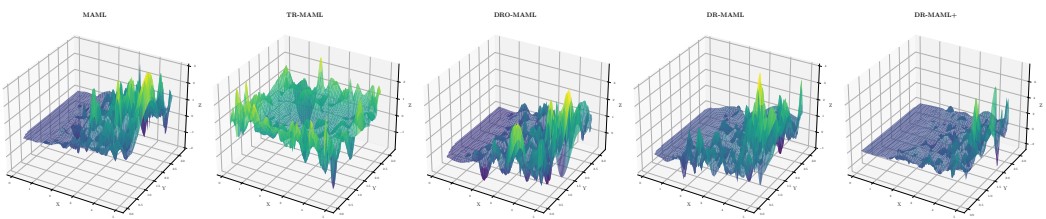

Figure 15: **The fast adaptation risk landscape of meta-trained MAML, TR-MAML, DRO-MAML, DR-MAML and DR-MAML+.** The figure illustrates a 5-shot sinusoid regression example, mapping to the function space $f(x) = A\sin(x - B)$. The $X$-axis and $Y$-axis represent the amplitude parameter $a$ and phase parameter $b$ respectively. The plots exhibit testing MSEs on the $Z$-axis across random trials of task generation.

### E.4 Further Exploration on Adaptation

We demonstrate the adaptation risk landscape of meta-trained MAML [25], TR-MAML [42], DRO-MAML [66], DR-MAML [1] and our DR-MAML+ in Fig. 15. The adaptation risk landscape shows the superiority of our method in optimizing within the expected tail risk minimization. Compared to other methods, DR-MAML+ exhibits smoother and smaller risk profiles, illustrating its robustness even in challenging tasks.

## F Computational Platforms & Softwares

This work employs Pytorch [85] as the default deep learning toolkit when implementing the developed methods. As for baselines, TR-MAMAL follows the standard implementation as work [42] and runs with Tensorflow [86]. Others are implemented with Pytorch. All experimental results are computed by NVIDIA RTX6000 GPUs and A800 GPUs.

