# OpenReview forum: "Theoretical Investigations and Practical Enhancements on Tail Task Risk Minimization in Meta Learning"
_NeurIPS.cc/2024/Conference — NeurIPS 2024 poster_

### Official Review · Reviewer_8vBx · 2024-07-08

**Soundness:** 4
**Presentation:** 4
**Contribution:** 3
**Rating:** 8
**Confidence:** 4

**Summary:**

This work investigates the tail risk minimization in meta-learning from theoretical and practical perspectives. Overall, this work is well-written, novel and theoretically enriches TR-MAML[1]/DR-MAML[2].


In the realm of large models, meta-learning plays a crucial role due to the pressing concern of distributional robustness across various tasks, particularly in risk-sensitive applications.
Here, I express my positive attitude based on this work's completeness, novelty, and workload.

[1] Collins, L., Mokhtari, A., & Shakkottai, S. (2020). Task-robust model-agnostic meta-learning. Advances in Neural Information Processing Systems, 33, 18860-18871.

[2] Wang, Q., Lv, Y., Xie, Z., & Huang, J. (2024). A simple yet effective strategy to robustify the meta learning paradigm. Advances in Neural Information Processing Systems, 36.

**Strengths:**

The novelty of this work lies in two aspects.

(1) This work reduces the distributionally robust meta-learning to a max-min optimization and performs analysis on tail risk generalization, convergence rate estimation, and the role of quantile estimation;

(2) This work enhances DR-MAML through more accurate quantile estimates with theoretical support.

Overall, this work clarifies its contribution in Table3 and completes all claims together with detailed proofs. Extensive experiments further verify theoretical insights.

**Weaknesses:**

No concrete weakness.

**Questions:**

With complete proofs and empirical evaluations, I do not have too many concrete questions except for some discussions.

(1) Both group DRO and tail risk minimization handle the distribution shift in robust optimization; what are the advantages or disadvantages between them in meta-learning?

(2) What difficulties will we encounter if we adopt the developed strategy in the optimization of large models?

---

> ### Author Rebuttal · Authors · 2024-08-06
>
> We sincerely thank **# Reviewer 8vBx** for these helpful comments. The remainders focus on questions to answer.
>
> ---
>
> **1. Advantages and disadvantages between group DRO and tail risk minimization in meta-learning**
>
> Thanks for this comment. The group DRO method employs risk reweighted algorithm to relax the weights of tasks and assign more weights to the gradient of worst cases. The tail risk minimization principle adopts two-stage optimization strategies to control the worst fast adaptation cases at a certain probabilistic level. (1) Group DRO is advantageous when predefined groups are available to guarantee robust performance across these groups, however, meta training is in a task episodic manner and weakens the applicability of group DRO. Tail risk minimization is more flexible, does not require clearly defined groups, and is suitable for risk-sensitive scenarios. (2) In the experiments, the tail risk minimization method consistently outperforms group DRO, demonstrating the advantages of the two-stage optimization strategy in improving robustness.
>
>
> **2. Application to large models**
>
> Thanks for this constructive comment. To answer the scalability question in large NNs, e.g., large models, we directly run experiments with large models.
>
> **Experimental setup:** CLIP [1] is a recent large vision-language model; hence, we employ "ViT-B/16"-based CLIP as the backbone to enable few-shot learning in the same way as MaPLe (N_CTX: 2, MAX_EPOCH: 30) [2], scaling to large NNs in evaluation. SGD is the default optimizer with LR 0.0035, and A6000 GPUs work for computations. We examine tail task risk minimization effectiveness on three large datasets. The class number split setup in datasets (num train/num val/num test) is TieredImageNet (351/97/160), ImagenetA [3] (128/32/40), and ImagenetSketch (640/160/200).
>
> **Result and analysis:** We'll include the **global rebuttal Figure15-17** and analysis below in the manuscript:
> >Results are reported in **Fig. 15-17**. DR-MaPLe and DR-MaPLe+ consistently outperform baselines across both average and ${\rm CVaR}_\alpha$ indicators in $\texttt{5-way 1-shot}$ cases, demonstrating the advantage of the two-stage strategy in enhancing the robustness of few-shot learning. DR-MaPLe+ achieves better results as KDE quantiles are more accurate with large batch sizes. These results examine the scalability and compatibility of our method on large models.
>
> **3. Opensource plan**
>
> Even though this work contributes to more theoretical points, we hope our empirical investigations can provide more insights in developing large model augmented few-shot learners.
>
> ***We'll opensource codes of large models' experiments to facilitate robust fast adaptation research from the updated manuscript.***
>
> ___
> **Reference**
>
> [1] Radford, Alec, et al. "Learning transferable visual models from natural language supervision." ICML 2021.
>
> [2] Khattak, Muhammad Uzair, et al. "Maple: Multi-modal prompt learning." CVPR 2023.
>
> [3] Hendrycks, Dan, et al. "Natural adversarial examples." CVPR 2021.
>
> ---
>
> *Finally, we hope your questions are well answered. And thanks for your suggestions in improving the quality of this manuscript.*
>
> ---

---

> > ### Comment · Reviewer_8vBx · 2024-08-08
> > **Thanks for the rebuttal and update the review**
> >
> > After reading the rebuttal and other reviewers' comments, I add further reviews.
> >
> > (1) Overall, this is a little niche but comprehensive theoretical paper about the tail risk minimization for meta learning.
> >
> > (2) The extra effort made by the authors is impressive in the era of large models. These new results on tail risk minimization for MaPLe are inspiring and well answer my lasting question about the role of CLIP like models in few-shot prediction. I am happy to see the benefits of tail risk minimization in large-model augmented meta-learning and looking forward to the open-sourced code.
> >
> > For the above, I thank the authors for the rebuttal and tend to raise my score as the bonus.

---

> > > ### Author Response · Authors · 2024-08-08
> > >
> > > Thank you for your helpful suggestions and kindness. Your comments help improve the manuscript a lot.

---

### Official Review · Reviewer_6Ej4 · 2024-07-09

**Soundness:** 3
**Presentation:** 3
**Contribution:** 3
**Rating:** 5
**Confidence:** 3

**Summary:**

This paper proposes an enhancement to the previous work termed DR-MAML by reformulating it as a Stackelberg game. Theoretical investigations regarding its solution concept, convergence rate, and generalization bound are provided. Numerical experiments demonstrate the improved robustness of the proposed method.

**Strengths:**

1. Both theoretical guarantees and numerical validations are provided to justify the sought robustness.
2. Reformulating DR-MAML's objective as a Stackelberg game enables analyzing the problem from the viewpoint of game theory.

**Weaknesses:**

1. The contribution of this work is incremental and exclusive to DR-MAML, which limits its scope and broader applicability.
2. The writing is hard to parse, and several notions are pretty vague. For instance, in lines 95-96, $F_{\ell}^{\alpha}$, $\Omega_{\alpha,\tau}$, and $p_{\alpha}$ are defined through illustration in a figure instead of mathematical expressions.
3. Experiments are limited and lack comparison to SOTA methods. It is recommended comparing DR-MAML+ with popular meta-learning methods (such as MetaCurvature and MetaOptNet) on open-source benchmark datasets including tieredImageNet, CUB-200-2011, and MetaDatasets. In addition, the improvement in Table 1 and Table 2 are rather marginal (no more than 0.6%).
4. There is no comparison of time and space complexity, and the scalability to large NNs like ResNet is unknown.
5. The last statement of Theorem 4.1 is imprecise. According to Eq. (22), this convergence rate solely holds true when $t$ approaches infinity.
6. The paper contains numerous typos. For instance, in line 4, the citation is not compiled. In line 95, "the resulling" should be corrected to "resulting." Additionally, in lines 96, 130, 240, 241, 244, 246, 254, 265, 283, 295, 303, 319, 328, 332, and 335-337, the cross-references are in the equation style instead of the correct format.

**Questions:**

See above.

**Limitations:**

No explicit discussion on limitations are provided.

---

> ### Author Rebuttal · Authors · 2024-08-06
>
> We sincerely thank **# Reviewer 6Ej4** for these insightful comments. The remainder mainly focuses on concerns to address.
>
> ---
>
> **1. Application scopes and contribution clarifications**
>
> Thanks for the comment. Sorry for confusing you in contributions, and we further summarize:
>
> (1) Regarding the application scopes, ***this work is meta-learning method agnostic***. Apart from MAML, CNP is also used in examinations. Besides, we've taken the advice to conduct experiments on more benchmarks with large models' augmented backbone. See **Point 3**.
>
> (2) Regarding the contribution, this work leans more on theoretical investigations of tail task risk minimization for meta-learning, which complements the empirical discoveries in Wang et al. (2023a). These include (i) the notion of solutions, (ii) understanding the Stackelberg game, (iii) generalization and asymptotic analysis in tail adaptation risk, and implementation tricks to enhance robustness. The practical enhancement is the theory's side product. See details in **Lines 35-39**, **Lines 602-604**, and **Table 3**.
>
> **2. More descriptions on notations**
>
> Thank you for your suggestions. **We'll add more to Line 96 as follows**:
>
> >The normalized cumulative distribution $F_{\ell}^{\alpha}(l;\theta)$ is defined as:
> $$F\_{\ell}^{\alpha}(l;\theta)=
> \begin{cases}
> 0, & l<\text{VaR}\_{\alpha}[\ell(\mathcal{T},\theta)]\ or \
> \frac{F\_{\ell}(l;\theta)-\alpha}{1-\alpha}, & l\geq\text{VaR}\_{\alpha}[\ell(\mathcal{T},\theta)].
> \end{cases}$$
>
> > $\forall\theta\in\Theta$, the meta learning operator $\mathcal{M}\_{\theta}$ defines:$\mathcal{M}\_{\theta}:\tau\mapsto\ell(\mathfrak{D}\_{\tau}^{Q},\mathfrak{D}\_{\tau}^{S};\theta).$ Accordingly, the tail risk task subspace $\Omega\_{\alpha,\tau}:=\bigcup\_{\ell\geq\text{VaR}\_{\alpha}[\ell(\mathcal{T},\theta)]}\left[\mathcal{M}\_{\theta}^{-1}(\ell)\right]$, with the task distribution constrained in $\Omega\_{\alpha,\tau}$ by $p_{\alpha}(\tau;\theta)$.
>
> **3. Additional experiments and larger NNs**
>
> Thanks for these precious suggestions.
>
> (1)  Comparison with MetaCurvature/MetaOptNet and more evaluation in larger NNs:
>
> Both [1] and our work are agnostic to meta-learning methods, with baseline selections specifically related to distributional robustness in meta-learning. Instead, we directly run experiments with large models. CLIP [2] is a recent SOTA than MetaCurvature/MetaOptNet; hence, we employ "ViT-B/16"-based CLIP as the backbone to enable few-shot learning in the same way as MaPLe (N_CTX: 2, MAX_EPOCH: 30) [3], scaling to large NNs in evaluation. SGD is the default optimizer with LR 0.0035, and A6000 GPUs work for computations. We examine tail task risk minimization effectiveness on three large datasets. The class number split setup in datasets (num train/num val/num test) is TieredImageNet (351/97/160), ImagenetA [4] (128/32/40), and ImagenetSketch (640/160/200).
>
> We'll include the **global rebuttal Figure15-17** and analysis below in the manuscript:
> >Results are reported in **Fig. 15-17**. DR-MaPLe and DR-MaPLe+ consistently outperform baselines across both average and ${\rm CVaR}_\alpha$ indicators in $\texttt{5-way 1-shot}$ cases, demonstrating the advantage of the two-stage strategy in enhancing the robustness of few-shot learning. DR-MaPLe+ achieves better results as KDE quantiles are more accurate with large batch sizes. These results examine the scalability and compatibility of our method on large models.
>
> ***We'll opensource codes to facilitate robust fast adaptation research from the updated paper.***
>
> (2) We've updated related work in **Section2** **Lines 61-63**:
> >Widely known are model agnostic meta learning and related variants, such as MetaCurvature, which learns curvature information and transforms gradients in the inner-loop optimization.
> >The metrics-based methods ... For example, MetaOptNet proposes to learn embeddings under a linear classifier and achieve SOTA few-shot classification performance.
>
>
> **Reference**
>
> [1] Wang, Qi, et al. "A simple yet effective strategy to robustify the meta learning paradigm." NeurIPS 2023.
>
> [2] Radford, Alec, et al. "Learning transferable visual models from natural language supervision." ICML 2021.
>
> [3] Khattak, Muhammad Uzair, et al. "Maple: Multi-modal prompt learning." CVPR 2023.
>
> [4] Hendrycks, Dan, et al. "Natural adversarial examples." CVPR 2021.
>
> **4. More analysis and rephrase**
>
> (1) Marginal improvement in **Table 1** and **Table 2**:
>
> Image processing necessitates a small batch size, causing smaller differences. See analysis in **Lines 305-307**:
> >... we attribute this to the small batch size in training, which weakens KDE's quantile approximation advantage.
>
> (2) Time and space complexity comparison:
>
> Sorry for missing this part. **We'll add the following to the manuscript:**
> >The space complexity is specific to the meta-learning method, while this work is agnostic to it. Hence, we report the computational complexity for the DR-MAML+ as $\mathcal{O}\big(\mathcal{B}\big(\mathcal{B} - \alpha|\mathcal{M}| \big)\big)$ while using KDE with the Gaussian kernel, and that of DR-MAML is $\mathcal{O}\big(\mathcal{B}\big(\log(\mathcal{B}) - \alpha|\mathcal{M}|\big)\big)$.
>
> (3) Refine theorem 4.1 condition:
>
> You are right, and **we’ll refine it as:**
> > Let the iteration sequence ... when $t$ approaches infinity.
>
> **5. Typos and equation-style reference**
>
> Sorry for this, we'll correct the "resullting" typo to "resulting" and modify all cross-reference styles, e.g., Fig/Table/Theorem/Assumption/Example, by removing unnecessary brackets "()", such as change "Fig. (6)" to "Fig. 6", "Theorem (4.2)" to "Theorem 4.2", "Assumption (1)" to "Assumption 1". We hope there is no typo and cross-reference style issues this time.
>
> ---
>
> *Thanks again for carefully reading our manuscript and proposing constructive suggestions. After clarifying the contribution and including more results, we hope the evaluation of this work can be reconsidered. Your help is precious to us.*

---

> > ### Comment · Reviewer_6Ej4 · 2024-08-11
> >
> > Thank you for providing the detailed response and additional experiments, which addressed most of my concerns.
> >
> > 1. Regarding the scope, could you please elaborate more on why this paper is "meta-learning method agnostic." The CVaR objective Eq. (3) of this work is specific to DR-MAML [1], and is not applicable to generic meta-learning approaches. Moreover, the theoretical investigations in Section 4 are also tailored to DR-MAML [1].
> >
> > 2. Could you please also provide the practical implementation time (throughput) and space (GPU memory) comparisons?

---

> ### Author Response · Authors · 2024-08-12
> **Thanks for feedback**
>
> We're happy to see most of the concerns are addressed, and we express sincere gratitude for the helpful feedback and suggestions. The following answers other questions:
>
> 1. We apologize for not clearly explaining that this work is agnostic to meta-learning methods.
>
> - Eq. (3) is not specifically for DR-MAML but is a general form of the risk function $\ell(\mathfrak{D}\_{\tau}^{Q},\mathfrak{D}\_{\tau}^{S};\theta)$ used in typical meta-learning methods. **Eq. (4)** is an example of Eq. (3) and is specific to DR-MAML. In this case, the risk function becomes $\ell(\mathfrak{D}\_{\tau}^{Q};\theta-\lambda\nabla\_{\theta}\ell(\mathfrak{D}\_{\tau}^{S};\theta))$, implying the implementation of MAML. **Algorithm 2 on Lines 661-662** shows the application of another method CNP, where the loss function is specifically $\ell(\mathfrak{D}\_{{\tau}}^{Q};z,\theta)$, with $z=h\_{\theta\_1}(\mathfrak{D}\_{\tau}^{S})$.
>
> - The theoretical insights in Section 4 are considered in a general form of meta learning with risk function $\ell(\mathfrak{D}\_{\tau}^{Q},\mathfrak{D}\_{\tau}^{S};\theta)$, not limited to MAML cases.
> Besides, we take MAML as a specific example for conducting theoretical analysis in Appendix Theorem C.1 on Lines 779-782.
>
> 2. The practical implementation time/space comparisons. We keep the setup the same as that in [1]:  use the same maximum number of meta gradient updates for all baselines in training processes (this means given $\alpha=0.5$, the tail risk minimization principle requires double task batches to evaluate and screen sub-batches).  For practical implementation, we take vanilla MaPLe, DR-MaPLe (MC augmented tail risk minimization), and DR-MaPLe+ (KDE augmented tail risk minimization) in tieredImageNet few-shot classification as the example to report. The Table below reports the overall training time and memory (the vanilla MaPLe serves as the anchor point, and + means additional costs from the two-stage operation).
>
> | -                        | MaPLe | DR-MaPLe | DR-MaPLe + |
> |--------------------------|------|---------|----------|
> | Implementation time      |   2.1h  |    +1.7h  |    +1.7h      |
> | Memory     |   41.57G  |    +36.84G  |    +36.84G      |
>
> Despite the more complex quantile estimations, DR-MaPLe+ does not show significantly higher time and space consumption compared to DR-MaPLe. It can be seen that both DR-MaPLe and DR-MaPLe+ consume more memories, and the extra training time over MaPLe arises from the evaluation and sub-batch screening in the forward pass. Such additional computations and memory costs bring significant robustness improvement, which can be crucial in risk-sensitive fast adaptation. We'll also include this point as a potential limitation in the Appendix.
>
> [1] Wang, Qi, et al. "A simple yet effective strategy to robustify the meta learning paradigm." NeurIPS 2023.
>
>
> ---
>
> *Feel free to let us know if these questions have been answered well, and we’re happy to engage in further discussions. We’ll incorporate these precious suggestions into the manuscript. It would be appreciated if you could update the scores after the concerns are addressed. Your help means a lot to us.*

---

> > ### Comment · Reviewer_6Ej4 · 2024-08-12
> >
> > Thank you for the elaborations and further experiments. I have no other concerns and I will update my rating.

---

> > > ### Author Response · Authors · 2024-08-12
> > > **Thanks for the update**
> > >
> > > Your suggestions help improve our manuscript a lot, and we express gratitude for your support.

---

### Official Review · Reviewer_ci3Y · 2024-07-23

**Soundness:** 3
**Presentation:** 2
**Contribution:** 3
**Rating:** 6
**Confidence:** 2

**Summary:**

The paper provides theoretical investigations for better understanding of an existing method in literature that focuses on minimizing expected tail risk. Equivalence of the algorithm is shown to a Stackerlberg game, which allows to study its convergence rate and asymptotic bounds on performance and generalization are provided. Finally, a practical heuristic is suggested to obtain performance improvements over the base method.

**Strengths:**

- The presentation and structure of the paper is good.
- The theoretical results, while limited in mathematical novelty, seem comprehensive and are clear to follow.
- Sufficient experiments and ablations are provided.

**Weaknesses:**

Clarity:
- It is not clear how the expression for the expected tail risk minimization is obtained in (3); the reviewer had to refer to the corresponding reference to understand it and thus, including it would help make the paper self-contained.
- Figure 1 is not clear; the font size needs to be increased.

**Questions:**

- What is the overhead induced by searching for the appropriate value of hyperparameter $\alpha$, compared to other baselines?

---

> ### Author Rebuttal · Authors · 2024-08-05
>
> We sincerely thank **# Reviewer ci3Y** for these insightful comments. The remainders focus on concerns and questions to address.
>
> ---
>
> **1. Additional explanations on the expected tail risk minimization**
>
> Thanks for this advice. **We'll include more explanations in Line 115** as follows:
> > Wang et al. [1] minimizes the expected tail risk, or equivalently $\text{CVaR}\_{\alpha}$ risk measure:
> $$
> \min\_{\theta\in\Theta}\mathcal{E}\_{\alpha}(\theta):=\mathbb{E}\_{p_{\alpha}(\tau;\theta)}
>         \Big[\ell(\mathfrak{D}\_{\tau}^{Q},\mathfrak{D}\_{\tau}^{S};\theta)
>         \Big].
> $$
> >Due to no closed form of $p\_{\alpha}(\tau;\theta)$, [1] introduces a slack variable $\xi\in\mathbb{R}$ and reformulates the objective as follows:
> $$
> \min_{\theta\in\Theta,\xi\in\mathbb{R}}
>         \mathcal{E}\_{\alpha}(\theta,\xi):=
>         \frac{1}{1-\alpha}\int\_{\alpha}^{1}v\_{\beta}d\beta
>         =
>         \xi+\frac{1}{1-\alpha}\mathbb{E}\_{p(\tau)}
>         \left[\left[\ell(\mathfrak{D}\_{\tau}^{Q},\mathfrak{D}\_{\tau}^{S};\theta)
>         -\xi\right]^{+}
>         \right],
> $$
> >where $v\_\beta:=F\_{\ell}^{-1}(\beta)$ denotes the quantile statistics and $\left[\ell(\mathfrak{D}\_{\tau}^{Q},\mathfrak{D}\_{\tau}^{S};\theta)-\xi\right]^{+}:=\max$ {$\ell(\mathfrak{D}\_{\tau}^{Q},\mathfrak{D}\_{\tau}^{S};\theta)-\xi,0$} is the hinge risk.
>
> We hope such descriptions make it easier to understand.
>
> **2. Larger font size in figure 1**
>
> Thanks for the valuable feedback. We've increased the font size in Figure 1 and will update it in the final manuscript.
>
> **3. Questions about hyperparameter $\alpha$**
>
> Thanks for your insightful question.
>
> (1) In our experiments, we did not perform extensive hyperparameter $\alpha$ searches. Instead, we adopted hyperparameter settings that are consistent with existing literature [1]. This ensures that our results are fair in comparison to previous studies without introducing additional overhead from hyperparameter tuning.
>
> (2) To further reveal the impact of confidence levels $\alpha$ on model performance, we conducted ablation experiments, as shown in **Fig. 12/13**. We can observe that in both sinusoid 5-shot and 10-shot tasks, as the confidence level varies, DR-MAML+ exhibits more stable performance than DR-MAML, indicating that DR-MAML+ has a lower sensitivity to confidence levels.
>
> ---
>
> **Reference**
>
> [1] Wang, Qi, et al. "A simple yet effective strategy to robustify the meta learning paradigm." NeurIPS 2023.
>
> ---
>
> *Finally, we hope these questions and concerns are well answered and addressed. Thanks again for your efforts. We are happy to discuss any other questions and provide further clarifications.*

---

> ### Comment · Reviewer_ci3Y · 2024-08-10
> **Acknowledgement of Rebuttal**
>
> I thank the authors for responding to the reviewers comments and feedback, and I appreciate the additional evaluation on larger models.
>
> However, I am inclined to agree with the other reviewers regarding the limited scope of the work. Additionally, the practical benefits of the proposed method is largely tangential to the main theoretical contribution about the Stackerlberg game equivalence, and arises from improving the risk estimates using more sophisticated density estimation, something that is already noted in the original work it builds from [1]. For these reasons, I keep my score as is.
>
> [1] Wang, Qi, et al. "A simple yet effective strategy to robustify the meta learning paradigm." NeurIPS 2023.

---

> ### Author Response · Authors · 2024-08-10
> **Thanks for precious feedback**
>
> Thanks for recognizing our primary theoretical contributions.
>
> ---
>
> You're right; the scope of this work is restricted to a theoretical understanding of tail risk minimization in meta learning.
>
> The practical benefits of quantile estimate are indeed inspired by the hypothesis in [1], and we (i) *establish rigorous theoretical analysis in* **Theorem 4.4** and **Remark 1**; (ii) *empirically conduct large-model few-shot learning experiments and verify the advantage of KDE over MC with larger batch size, complementing the theory*.
>
> Ref: [1] Wang, Qi, et al. "A simple yet effective strategy to robustify the meta learning paradigm." NeurIPS 2023.
>
> ---
>
> *Finally, we're encouraged by your positive comments, and your valuable suggestions help improve the manuscript greatly. Many thanks.*

---

### Author Rebuttal · Authors · 2024-08-06

*We sincerely thank all reviewers and area chairs for their work. This global response summarizes reviews, addresses concerns, answers questions, and reports changes in the manuscript.*

---

### **I. Review Summary**

We thank all reviewers for their comments:
1. a *well-constructed paper easy to follow* **[# Reviewer ci3Y and # Reviewer 8vBx]**;
2. *comprehensive theoretical analysis aligned with numerical evaluations* **[# Reviewer ci3Y, # Reviewer 6Ej4, and # Reviewer 8vBx]**;
3. *addresses the problem from Stackelberg game/max-min optimization* **[# Reviewer 6Ej4 and # Reviewer 8vBx]**.

---

### **II. Primary Concerns and Questions**

**1. Contributions and application scope clarifications [# Reviewer 6Ej4]**

(1) Regarding the contribution, this work leans more on theoretical points **[# Reviewer ci3Y and # Reviewer 8vBx]** throughout tail risk minimization for meta-learning, which complements the empirical discoveries in Wang et al. (2023a) [1]. These include (i) notion of solutions, (ii) algorithmic understanding from the Stackelberg game, (iii) generalization, asymptotic analysis in tail adaptation risk, quantile estimates' influence, and implementation tricks to enhance robustness as mentioned in **Lines 35-39** and **Table 3**. The practical enhancement is the side product of the theory.

(2) Regarding the application scopes, *this work is meta-learning method agnostic*. Apart from MAML, CNP is also used in examinations. Importantly, we've taken the advice to ***conduct experiments on more benchmarks with large models' augmented backbone***. See **Point 2** and detailed **Response to # Reviewers 6Ej4/8vBx**.

**2. Comparison with MetaCurvature/MetaOptNet and more evaluation [# Reviewer 6Ej4], Scalability in large models [# Reviewer 6Ej4 and # Reviewer 8vBx]**

Both [1] and our work are agnostic to meta-learning methods, with baseline selections specifically related to distributional robustness in meta-learning. Instead, we directly run experiments with large models. CLIP [2] is a recent SOTA than MetaCurvature/MetaOptNet; hence, we employ "ViT-B/16"-based CLIP as the backbone to enable few-shot learning in the same way as MaPLe (N_CTX: 2, MAX_EPOCH: 30) [3], scaling to large NNs in evaluation. SGD is the default optimizer with LR 0.0035, and A6000 GPUs work for computations. We examine tail task risk minimization effectiveness on three large datasets. The class number split setup in datasets (num train/num val/num test) is TieredImageNet (351/97/160), ImagenetA [4] (128/32/40), and ImagenetSketch (640/160/200).

We've attached the results in **Figure15-17** and analysis below:
>Results are reported in **Fig. 15-17**. DR-MaPLe and DR-MaPLe+ consistently outperform baselines across both average and ${\rm CVaR}_\alpha$ indicators in $\texttt{5-way 1-shot}$ cases, demonstrating the advantage of the two-stage strategy in enhancing the robustness of few-shot learning. DR-MaPLe+ achieves better results as KDE quantiles are more accurate with large batch sizes. These results examine the scalability and compatibility of our method on large models.

**3. Eq. (3) descriptions, larger font size in Fig. 1, more analysis, one typo and cross-reference style [# Reviewer ci3Y and # Reviewer 6Ej4]**

(1)	We've added Eq. (3) and other notation descriptions. See individual responses.
(2)	We've enlarged Fig. 1's font size in the updated version.
(3)	We've refined Theorem 4.1 to make our statement more precise, and time/space complexity analysis sees **Response to # Reviewer 6Ej4**.
(4) Typos and equation-style reference:
We've corrected the “resullting” typo to “resulting” and modified all cross-reference styles, e.g., Fig/Table/Theorem/Assumption/Example, by removing unnecessary brackets “()”, such as changing “Fig. (6)” to “Fig. 6”, “Theorem (4.2)” to “Theorem 4.2”, “Assumption (1)” to “Assumption 1”. We've proofread the manuscript many times, securing no typos and other informal reference style uses.

**4. Other questions and opensource plan**

Thanks and we leave other questions and answers in individual responses. *We'll opensource codes of large models' experiments to facilitate robust fast adaptation research in the updated manuscript.*

**Reference**

[1] Wang, Qi, et al. "A simple yet effective strategy to robustify the meta learning paradigm." NeurIPS 2023.

[2] Radford, Alec, et al. "Learning transferable visual models from natural language supervision." ICML 2021.

[3] Khattak, Muhammad Uzair, et al. "Maple: Multi-modal prompt learning." CVPR 2023.

[4] Hendrycks, Dan, et al. "Natural adversarial examples." CVPR 2021.

---
### **III. Manuscript Future Changes**

**1. Include new experimental results with large model backbones.**

**2. Add descriptions and discussions, and revise cross-reference styles as previously mentioned.**

---

*Once again, thank you to all the reviewers and area chairs. Your effort means a lot in improving our manuscript.*

---

### Decision · Program_Chairs · 2024-09-25

**Decision:**

Accept (poster)

**Comment:**

This paper investigates tail risk minimization in meta-learning, providing theoretical analysis and a practical enhancement to the DR-MAML method. Reviewers generally found the paper technically sound with a good presentation and valuable theoretical contributions, including the Stackelberg game equivalence, convergence rate analysis, and generalization bounds.
There were few initial concerns regarding the limited scope and incremental nature of the contribution (6Ej4), the fact that the practical benefits were also seen as tangential to the main theoretical contribution (ci3Y), and limited experimental evaluation that lacking comparisons to SOTA methods and larger models (6Ej4). These were subsequently addressed by authors during rebuttal stage by clarifying the broader applicability of their theoretical analysis and providing additional analysis / experiments.
Overall, The paper presents a solid theoretical investigation of tail risk minimization in meta-learning and demonstrates its practical benefits. The authors effectively addressed initial concerns regarding scope, evaluation, and applicability to large models, leading to a consensus among reviewers on acceptance.